# TRQA: Time Series Reasoning Question And Answering Benchmark

## Abstract

Time series data underpin critical applications across domains such as finance, healthcare, transportation, and environmental science. While recent work has begun to explore multi-task time series question answering (QA), current benchmarks remain limited in scope, with an emphasis largely on forecasting and anomaly detection tasks. We introduce TRQA, a novel time series QA benchmark that substantially broadens task coverage and provides a unified setting for evaluating diverse temporal reasoning abilities. TRQA unifies six diverse tasks under a single framework, organized into two complementary groups: (1) *conventional reasoning tasks*, including anomaly detection and classification, and (2) *advanced reasoning tasks*, such as characterization, comparison, data transformation, and temporal relationship reasoning. These tasks span multiple question types, such as *true-or-false (TF)*, *multiple-choice (MC)*, and a novel *puzzling (PZ)*, enabling a more comprehensive evaluation of diverse aspects of time series reasoning. We curated a large-scale dataset with 210k samples, covering a diverse 13 domains, 6 tasks, and 3 types of questions. Each sample consists of one or more time series, an accompanying question, contextual information about the time series, and a corresponding answer. Zero-shot evaluation demonstrates that these tasks are challenging for both commercial and open-source Large Language Models (LLMs). For example, the best-performing commercial LLM, Gemini-2.5-Flash, achieves an average score of only 65.08. While open-source LLMs show notable performance gains after instruction tuning, there remains considerale room for improvement. For instance, the best-performing open-source model, LLaMA-3.1-8B, reaches an average score of 85.26, suggesting that these tasks are still non-trivial and pose ongoing challenges for current models. The data are available in GitHub: `https://anonymous.4open.science/r/TRQA_benchmark-6737`.

## 1 Introduction

Time series data are ubiquitous, arising naturally in domains such as financial markets, electronic health records, environmental monitoring, and energy management. Effectively reasoning over temporal patterns is therefore essential for real-world decision-making. Traditionally, research in time series has concentrated on a relatively narrow set of tasks, most notably forecasting future values, anomaly detection, imputation, and classification (Torres et al., 2021; Lim & Zohren, 2021; Wen et al., 2022). While these problems are fundamental and have important applications, the scope of temporal reasoning extends far beyond these settings, encompassing a richer set of queries that demand deeper understanding of fundamental characteristics and patterns of time series.

Recent advances in Large Language Models (LLMs) have revolutionized natural language processing and multimodal learning, demonstrating remarkable capabilities in understanding, reasoning, and generating across diverse domains (OpenAI et al., 2024; Grattafiori et al., 2024; Team et al., 2025b; Yang et al., 2025). This progress has inspired a growing interest in applying LLMs to time series analysis. Early studies have explored leveraging LLMs for classical time series tasks, such as forecasting and anomaly detection (Zeng et al., 2023; Jin et al., 2023; Zhou & Yu, 2024; Zhang et al., 2024), often by converting temporal data into textual descriptions or prompt-based representations. However, most existing approaches primarily focus on numeric prediction, leaving open the

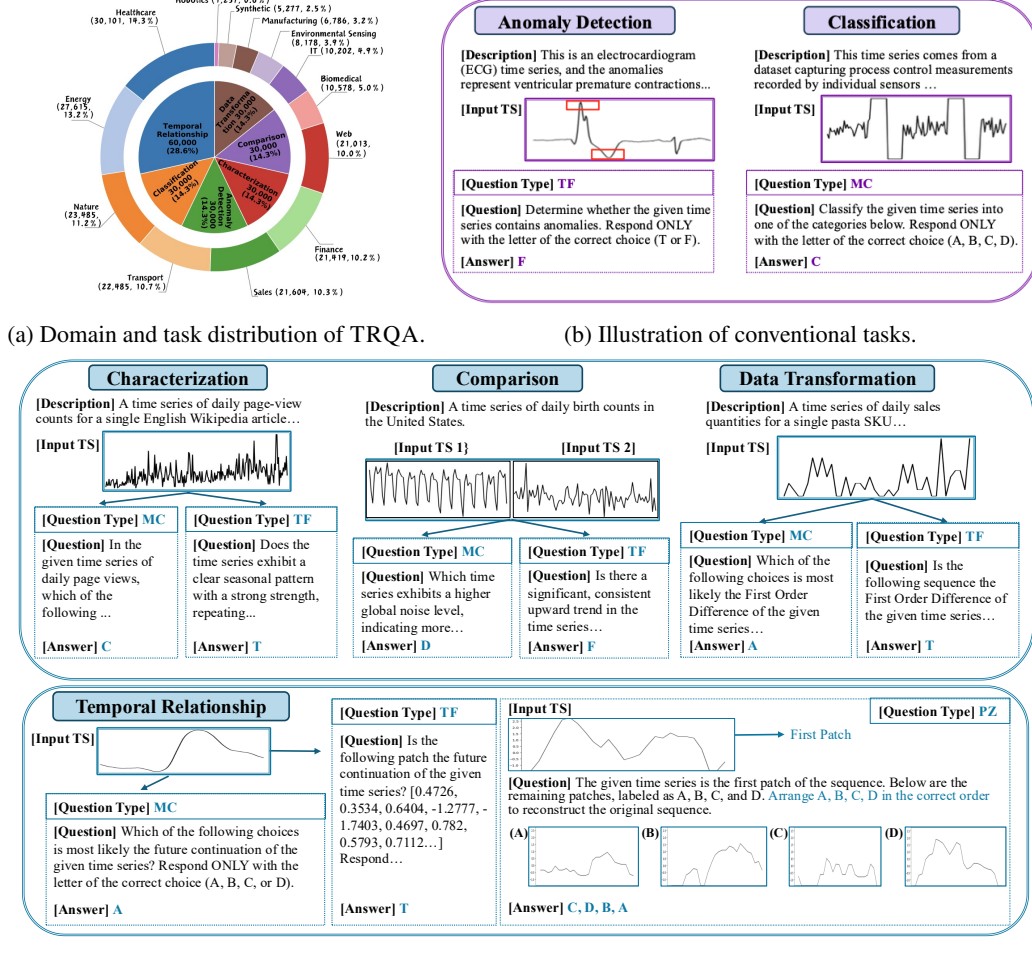

(a) Domain and task distribution of TRQA.

(b) Illustration of conventional tasks.

(c) Illustration of advanced tasks.

Figure 1: Data distribution and tasks of TRQA.

question of whether LLMs can develop stronger temporal reasoning abilities, such as understanding contextual information, and relationships across multiple time series.

Time series question answering (QA) has recently emerged as a promising paradigm for pushing the boundaries of time series modeling beyond traditional tasks (Merrill et al., 2024; Uddin et al., 2025; Xu et al., 2025a; Zhong et al., 2025a; Wang et al., 2025a; Kong et al., 2025). Rather than being limited to traditional tasks like forecasting, time series QA reformulates time series tasks through natural language queries, enabling models to tackle richer and more complex questions about temporal patterns and dynamics. This paradigm opens the door to evaluating a model's reasoning capabilities, such as understanding intrinsic characteristics of time series and identifying relationships across multiple sequences. For example, ChatTS (Xie et al., 2025) generates situational questions based on *synthetic* time-series attributes. ITFormer (Wang et al., 2025b) introduces EngineMT-QA, which is a *domain-specific* dataset for aero engine time series. Mtbench (Chen et al., 2025) proposes a QA benchmark mainly for *forecasting* tasks. Time-MQA (Kong et al., 2025) constructs question–answer pairs that span both numeric reasoning tasks and open-ended QA tasks, but its *open-ended* answers are difficult to evaluate objectively. While these efforts mark important progress, they are constrained by synthetic or domain-specific data and narrowly scoped tasks. Moreover, some evaluation protocols—particularly for open-ended answers—remain difficult to standardize, limiting fair comparison across models.

In this paper, we introduce TRQA, a large-scale benchmark that addresses these limitations by covering diverse domains and tasks, while also providing standardized evaluation protocols. A direct

Table 1: Comparison of time series question answering datasets and benchmarks.

| Dataset | Tasks Scope | # Reason Tasks | # Question Type | Human Eval | # Domain | Size |
|---|---|---|---|---|---|---|
| TS-Insights (Zhang et al., 2023) | Captioning | 1 | 1 | ✗ | 7 | 100k |
| TSandLanguage (Merrill et al., 2024) | Forecasting | 3 | 2 | ✓ | 10 | 230k |
| CiK (Williams et al., 2025) | Forecasting | 1 | 1 | ✓ | 7 | 2.9k |
| MTBench (Chen et al., 2025) | Forecasting | 4 | 3 | ✗ | 2 | 42k |
| TimeSeriesExam (Cai et al., 2024) | Various | 5 | 1 | ✓ | 1 | 0.7k |
| ChatTS (Xie et al., 2025) | Various | 4 | 5 | ✓ | 4 | 2.2k |
| ITFormer (Wang et al., 2025b) | Various | 4 | 2 | ✓ | 1 | 11k |
| Time-MQA (Kong et al., 2025) | Various | 5 | 4 | ✗ | 12 | 200k |
| **TRQA** (ours) | Various | **6** | **3** | ✓ | **13** | **210k** |

comparison between TRQA and existing datasets is provided in Table 1. We curate and annotate 210k high-quality samples from 13 domains, as shown in Figure 1a. TRQA integrates 6 distinct tasks grouped into two complementary categories: (1) *Conventional Tasks*: *anomaly detection* and *classification*. (2) *Advanced Tasks*: *characterization*, *comparison*, *data transformation*, and *temporal relationship* reasoning. All tasks are cast into a unified QA format with three question types: *true-or-false (TF)*, *multiple-choice (MC)*, and a novel *puzzling (PZ)*. PZ questions resemble human-like problem settings (Fissler et al., 2018) and, as evidenced in computer vision (Noroozi & Favaro, 2016), provide a strong probe of general cognition. For time series, they are particularly valuable in evaluating models' ability to reason about temporal order and relational structure. Beyond task design, we carefully detail the *data collection process*, *benchmark construction*, *dataset statistics*, and *evaluation protocol*, ensuring rigorous transparency and reproducibility. Our benchmark provides a standardized platform to evaluate various LLMs (OpenAI et al., 2024; cla; Team et al., 2025b; Grattafiori et al., 2024; Yang et al., 2025). Initial empirical studies demonstrate that existing models struggle across several tasks, particularly in structural and relational reasoning, highlighting substantial future directions for improvement.

In summary, our main contributions are threefold. (1) We introduce TRQA, a novel large-scale benchmark comprising 210k samples across 13 domains, covering 6 tasks and 3 types of questions. (2) We provide a detailed description of the benchmark's construction along with comprehensive statistics. (3) We conduct extensive evaluations of TRQA using a wide range of popular commercial and open-source LLMs, accompanied by an in-depth analysis of their performance.

## 2 RELATED WORK

**Time Series Analysis: From Numbers to Narratives.** Traditional research on time series has primarily focused on numerical sequences, enabling core tasks such as forecasting (Torres et al., 2021), imputation (Wang et al., 2024), and classification (Mohammadi Foumani et al., 2024), often treating them as isolated numeric signals (Hamilton, 2020). In practice, however, time series are rarely independent of their surrounding context. They frequently interact with external information—such as textual reports, domain expertise, or heterogeneous side signals—that shapes or enriches their dynamics (Jiang et al., 2025; Xu et al., 2025b; Liu et al., 2025; 2024; Li et al., 2025). Recognizing this, recent work has moved beyond purely numeric modeling to incorporate multimodal signals across domains including healthcare (Johnson et al., 2016; 2023), finance (Li et al., 2024a; Dong et al., 2024), retail (Skenderi et al., 2024), and transportation (Li et al., 2024b). While much of this research leverages external modalities to improve numeric predictions on predefined tasks, a growing body of work instead positions *natural language as a richer interface* for time series, using language as the medium for querying, reasoning, and interpreting temporal patterns (Merrill et al., 2024; Williams et al., 2025; Wang et al., 2025b; Chen et al., 2025; Xie et al., 2025; Kong et al., 2025). Together, these efforts define the emerging direction of *time series question answering*.

**Large Models on Time Series.** Advances in large language models (LLMs) (Vaswani et al., 2017) have recently enabled general question answering over time series. A growing line of work integrates LLMs with time series for downstream tasks (Chang et al., 2023; Alnegheimish et al., 2024; Yu et al., 2023; Jin et al., 2023), with extensions to multimodal language models as well (Zhong et al., 2025b; Merrill et al., 2024; Moon et al., 2022). Given their strong generalization ability through natural

Table 2: Tasks of TRQA. TF, MC, and PZ denote true-or-false, multiple-choice, and puzzling.

| Group | Task | Description | Question Type |
|---|---|---|---|
| Conventional Tasks | Anomaly Detection | Determine whether the input contains anomalies. | TF |
| | Classification | Classify the input time series. | MC |
| Advanced Tasks | Characterization | Determine the characteristics of the time series. | TF & MC |
| | Comparison | Compare the characteristics of two time series. | TF & MC |
| | Data Transformation | Identify the relationship between raw and transformed data. | TF & MC |
| | Temporal Relationship | Determine the temporal relationship of patches. | TF & MC & PZ |

language interfaces, comprehensive evaluation is critical to ensure the transparency and reliability of LLMs in time series applications.

## 3 TRQA BENCHMARK

In this section, we introduce the proposed TRQA benchmark, which is designed to provide a benchmark for time series question answering. We begin by formulating the tasks and defining question types in Section 3.1. Next, Section 3.2 describes the data sources and preprocessing procedures. Section 3.3 then details the construction of the benchmark, including its structure and design considerations. Data statistics are discussed in Section 3.4. Finally, Section 3.5 outlines the evaluation protocols used to assess model performance.

### 3.1 TASK FORMULATION

**Task Taxonomy.** As shown in Table 2 and Figure 1, the proposed TRQA benchmark encompasses two groups of tasks with six diverse tasks designed to evaluate a model's ability of understanding the fundamental properties of time series data. The first group, *Conventional Tasks*, is comprised of the tasks widely studied in traditional time series analysis: (1) *Anomaly Detection*, identifying irregular or unexpected patterns in time series; (2) *Classification*, reasoning about the relationship between a time series and its underlying conceptual category. The second group, *Advanced Tasks*, consists of novel analytical tasks about intrinsic properties of time series: (3) *Characterization*, inferring fundamental properties such as trend, seasonality, and dispersion; (4) *Comparison*, reasoning about relative similarities and differences between two time series; (5) *Data Transformation*, understanding relationships between original and transformed time series, e.g., Fourier transform; and (6) *Temporal Relationship*, capturing the chronological dependencies among time series patches. These advanced tasks push the boundaries of conventional time series modeling, fostering the development of models that can grasp cognitive concepts of time series and reason over human questions.

To bring all tasks under a single umbrella, we formulate them in a unified Question-Answering (QA) format. Every instance is converted into a time series input $X$ paired with contextual information $C$ and a question $Q$, and the model is expected to provide the correct answer $A$, where $C$ and $Q$ are expressed by natural language. Let $f$ denote the model, then the TRQA problem is formulated as:

$$A = f(X, C, Q). \tag{1}$$

**Question Types.** Our TRQA benchmark encompasses a wide variety of question types, such as *true-or-false (TF)*, *multiple-choice (MC)*, and *puzzling (PZ)* questions. A TF question requires the model to determine whether a claim about the input time series is True (T) or False (F). A MC question requires the model to select the correct claim about the input. In addition, we introduce a novel *puzzling (PZ)* question to the time series QA community. PZ questions are valuable because they represent realistic, human-like problem settings Fissler et al. (2018) and have been shown to effectively evaluate models' general cognitive abilities, as demonstrated in computer vision Noroozi & Favaro (2016). In this question, the model is given the first patch of a time series, along with the remaining shuffled patches, and tasked with arranging them in the correct chronological order.

### 3.2 DATA COLLECTION

To construct the TRQA benchmark, we collect and preprocess time series data from diverse public sources, spanning domains such as healthcare, transportation, and finance, to ensure broad coverage

and representativeness. At its center are the core datasets, which serve as the primary foundation for a wide range of tasks. In addition, the benchmark integrates two specialized sources: classification datasets and anomaly detection datasets. This subsection describes these data sources and the selection criteria. More details can be found in Appendix B.

**Core Datasets.** We collect high-quality real-world time series data from a wide range of domains, including energy, finance, healthcare, nature, sales, transport, and web, which are used by time series foundation model benchmarks, such as Lotsa (Woo et al., 2024), Time-300B (Shi et al., 2024), and UTSD (Ma et al., 2024). To ensure data quality, we retain only sequences with a minimum length of 1k. We further filter sequences with a missing rate greater than 1% or an outlier rate (the proportion of points lying beyond three times the interquartile range (3×IQR) exceeding 5%). For each dataset, we refer to the original source to gather background information about the time series and provide a concise, one-sentence description. More details are presented in Appendix B.1

**Anomaly Detection Datasets.** We extract data from multiple time-series anomaly detection benchmarks (Paparrizos et al., 2022; Su et al., 2019), including ECG (Moody & Mark, 2001), SMD (Su et al., 2019), MGAB (Thill et al.) Genesis (von Birgelen & Niggemann, 2018), GHL (Filonov et al., 2016), Occupancy (Candanedo & Feldheim, 2016). These datasets span various domains, including healthcare, mathematical biology, spacecraft telemetry, industrial control systems, environmental sensing, cybersecurity on IT Operations. For each dataset, we summarize its description and domain information directly from the original papers. More details are presented in Appendix B.2.

**Classification Datasets.** Our classification data comes from the univariate UCR Archive (Dau et al., 2019a). We select datasets with at most four classes and sequence lengths under 400, and enrich them with textual descriptions from the official documentation. The resulting subset spans diverse domains, including robotics, energy, healthcare, synthetic, manufacturing (see Appendix B.3).

## 3.3 Benchmark Construction

In this subsection, we describe the construction of the benchmark for each task. To maintain balance across tasks, we allocate an equal number of samples (30k) to each task, except for temporal relationship, which we allocate 60k samples since PZ is very challenging. Except for classification and anomaly detection, samples for all other tasks are drawn from the *Core Datasets* (Section 3.2) using *Hierarchical Random Sampling* (Algorithm 1) to ensure a balanced distribution across domains, datasets, and sequences. Unless otherwise specified, all samples have a random length in $[32, 256]$, and are z-scored to reduce data bias. The term data bias refers specifically to scale-based shortcuts or magnitude variance across heterogeneous domains, rather than semantic or sampling bias (Appendix C.2). Finally, each task's samples are randomly partitioned into 70% for training, 10% for validation, and 20% for testing. The remainder of this section describes the construction process for each task. More details and examples can be found in Appendix C-D.

**Characterization.** The characterization task assesses the model's capability to reason about fundamental properties of time series, including trend, seasonality, and dispersion. Questions are posed as TF or MC, and final answers are determined through multi-LLM consensus.

Given a sample $\mathbf{x}$ and its meta data, we first instruct GPT-4o (Hurst et al., 2024) to generate question and answer pairs based on a randomly selected subset of one to three topics, and question type (TF or MC). Briefly, the process involves the following steps: (1) We instruct GPT to generate captions for the input and randomly select a sub-topic for each topic (e.g., selecting the sub-topic "trend direction" under the topic "trend"); (2) GPT is instructed to generate a QA pair based on the inputs, captions, sub-topics, and the specified question type; (3) GPT performs a self-check of the generated QA pair and provides a confidence score, where only QA pair with a high confidence is retained; (4) We further leverage other powerful LLMs, including GPT-4.1, Gemini-2.5-Flash, and Claude-3.5-Sonnet, along with the answer given by GPT-4o to produce a consensus answer and reduce model bias. For more details, please refer to Appendix C.3.

**Comparison.** The comparison task assesses the model's ability to reason about the relative characteristics of the two time series, such as shape and correlation. Similar to the characterization task, this task is also formulated as TF or MC questions. We first obtain an anchor sample $\mathbf{x}$ from domain $M$, dataset $D$, and sequence $S$. Given the anchor $\mathbf{x}$, we construct a set of 10 comparison samples $\{\mathbf{x}'_1 \dots \mathbf{x}'_{10}\}$ with the same length as $\mathbf{x}$. Among which, one is drawn from the sequence $S$, two from

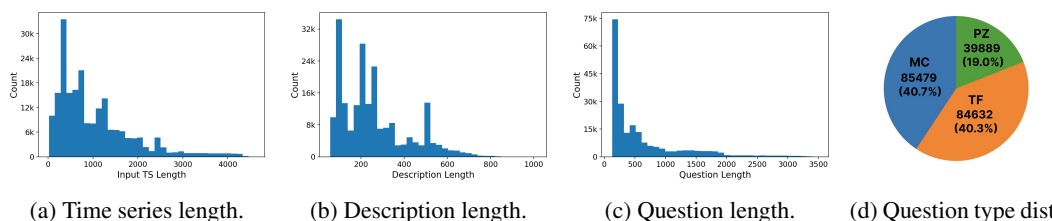

(a) Time series length.   (b) Description length.   (c) Question length.   (d) Question type dist.

Figure 2: Histograms of time series, description and question lengths, and question type distribution.

other sequences within dataset $D$, three from other datasets within domain $M$, and four from other domains. We also use a process similar to the characterization task to generate QA pairs. More details can be found in Appendix C.4.

**Data Transformation.** The data transformation task evaluates the model's ability to infer the transformation relationship between the input time series and its transformed counterpart, which is generated from the Fourier transform, wavelet transform, or first-order differencing. We then use predefined templates to formulate the task as either TF or MC questions. In TF questions, the model is asked to determine whether a given sequence is the correct transformation (e.g., the results of the Fourier transform) of the input time series $\mathbf{x}$. In MC questions, the model is required to select the correct transformed sequence given the input time series $\mathbf{x}$ and the specified transform operation (e.g., Fourier transform). All transformations are computed using professional libraries (Harris et al., 2020; Virtanen et al., 2020). The correct transformation is computed directly from the input $\mathbf{x}$, whereas incorrect transformations are generated from other randomly sampled time series $\mathbf{x}'$. For more details, please refer to Appendix C.5.

**Temporal Relationship.** The temporal relationship task evaluates the model's ability to infer the temporal structure among time series patches, testing 3 core reasoning capabilities: *Structural Continuity*, *Chronological Reasoning*, and *Contextual Discrimination*. This task is formulated as TF, MC, or PZ questions. Given the first chronological patch $\mathbf{x}$, a TF question asks the model to determine whether a candidate patch $\mathbf{y}$ is the immediate successor of $\mathbf{x}$, while an MC question asks the model to choose the correct next patch from candidates $[\mathbf{y}_1, \mathbf{y}_2, \mathbf{y}_3, \mathbf{y}_4]$. The false candidates are randomly sampled from the full dataset, but from sequences different from that of $\mathbf{x}$. A puzzling question presents four shuffled successor patches of $\mathbf{x}$ and asks the model to arrange them in the correct chronological order. All questions are generated using predefined templates. See Appendix C.6 for more details.

**Anomaly Detection.** The anomaly detection task evaluates the model's ability to recognize anomalous patterns in the input time series, which is formulated as a TF question. Each sample $\mathbf{x}$ is randomly cropped from a full sequence of the anomaly detection dataset. Since anomalous samples are much fewer than normal ones, we downsample the normal samples to balance the classes at a 1:1 ratio. The questions are composed using a predefined template. See Appendix C.7 for more details.

**Classification.** The classification task evaluates the model's ability to categorize input time series based on their patterns and characteristics. We reformulate the classification task into the MC question format, where the original numeric class, labels, e.g., 0 or 1, are converted into informative textual choices, e.g., "Cabernet Sauvignon" or "Shiraz". See Appendix C.8 for more details.

### 3.4 DATA STATISTICS

Figure 1a shows the distribution of domains and tasks, and Figure 2d shows the distribution of question types. Samples are nearly balanced across tasks, question types, and major domains. Figures 2a–2c present the histograms of time series length, description length, and question length, all of which exhibit long-tail distributions.

### 3.5 EVALUATION PROTOCOL

The TRQA benchmark includes three question types, each with a specific evaluation metric. TF and MC questions are evaluated using accuracy. PZ questions are scored by comparing each predicted

Table 3: Main results. A.D. denotes anomaly detection, CLS denotes classification. MC, TF, and PZ denote multiple-choice, true-or-false, and puzzling, respectively. SFT stands for supervised fine-tuning. The best and second-best results are highlighted in **bold** and underlined, respectively.

| Group | Task | A.D. | CLS | Characterization | | Comparison | | Data Transform | | Temporal Relation | | | Overall |
|---|---|---|---|---|---|---|---|---|---|---|---|---|---|
| | Question Type | MC | MC | TF | MC | TF | MC | TF | MC | TF | MC | PZ | |
| Zero Shot | GPT-4.1 | 55.85 | 50.38 | **92.97** | **89.36** | 83.57 | 76.99 | 54.36 | 51.13 | 65.90 | 79.09 | 45.77 | 62.82 |
| | GPT-4o | 54.32 | 47.20 | 88.15 | 84.15 | 78.61 | 69.07 | 60.66 | 53.24 | 62.25 | 75.58 | 45.61 | 60.73 |
| | Claude-3.5-Sonnet | 51.27 | 41.23 | 74.39 | 78.45 | 66.59 | 74.14 | 65.79 | 57.07 | 82.05 | 82.15 | 54.56 | 61.19 |
| | Gemini-2.5-Flash | 52.08 | 49.07 | 85.48 | 81.08 | 77.79 | 72.21 | 63.62 | 60.17 | 75.05 | 84.49 | 60.84 | 65.08 |
| | Qwen3-8B | 50.60 | 50.52 | 77.35 | 66.87 | 71.04 | 63.21 | 52.43 | 34.46 | 65.22 | 67.14 | 21.93 | 51.04 |
| | LLaMA3.1-8B | 54.92 | 50.20 | 68.10 | 62.26 | 67.84 | 49.98 | 51.90 | 36.56 | 54.82 | 40.95 | 6.80 | 44.93 |
| | Ministral-8B | 53.35 | 34.08 | 71.06 | 63.93 | 47.54 | 52.90 | 50.70 | 25.28 | 50.58 | 33.88 | 30.77 | 44.65 |
| | Qwen3-0.6B | 50.40 | 35.83 | 62.00 | 48.78 | 58.03 | 37.51 | 49.03 | 23.62 | 51.99 | 37.33 | 13.38 | 39.06 |
| | LLaMA3.2-1B | 49.47 | 39.48 | 63.74 | 52.55 | 61.02 | 36.82 | 48.87 | 4.20 | 48.97 | 5.44 | 6.76 | 35.70 |
| | Gemma3-1B | 49.15 | 49.83 | 63.74 | 47.71 | 61.19 | 43.37 | 49.37 | 24.88 | 49.42 | 25.84 | 23.97 | 43.03 |
| Instruction Tuning | Qwen3-8B | 87.70 | 90.05 | 92.37 | 85.42 | 86.55 | 79.08 | 89.84 | 84.99 | 96.84 | **97.56** | 66.21 | 84.29 |
| | LLaMA3.1-8B | **91.02** | **91.27** | 92.44 | 83.68 | **86.72** | **79.31** | 90.17 | **86.62** | 96.94 | 97.41 | **67.68** | **85.26** |
| | Ministral-8B | 71.56 | 74.28 | 91.31 | 80.78 | 84.14 | 74.63 | 75.15 | 71.61 | 94.07 | 94.15 | 56.82 | 74.74 |
| | Qwen3-0.6B | 83.68 | 85.78 | 89.38 | 74.87 | 80.65 | 64.84 | 80.51 | 73.28 | 93.92 | 93.79 | 63.34 | 78.32 |
| | LLaMA3.2-1B | 83.08 | 83.83 | 87.71 | 74.37 | 78.61 | 60.88 | 68.09 | 51.67 | 91.39 | 88.81 | 57.53 | 73.48 |
| | Gemma3-1B | 83.10 | 84.05 | 87.88 | 72.54 | 78.61 | 59.31 | 64.06 | 45.23 | 91.00 | 88.05 | 42.92 | 69.70 |

position with the ground truth and computing the proportion of correct matches. For example, with a ground truth A, B, C, D and prediction B, A, C, D, only the last two match, yielding 50% accuracy.

# 4 EXPERIMENTS

In this section, we present experimental results of the commercial LLMs and open-source LLMs on our TRQA benchmark, and provide analysis of the results.

## 4.1 MAIN RESULTS

We evaluate *zero-shot* performance of (1) commercial LLMs: GPT-4.1, GPT-4o, Claude-3.5-Sonnet and Gemini-2.5-Flash; (2) medium size open-source LLMs: Qwen3-8B (Yang et al., 2025), LLaMA3.1-8B (Dubey et al., 2024), Ministral-8B; (3) small size open-source LLMs: Qwen3-0.6B (Yang et al., 2025), LLaMA3.2-1B (Dubey et al., 2024), Gemma3-1B (Team et al., 2025a). We further apply *instruction tuning* Peng et al. (2023) to the open-source methods using LoRA Hu et al. (2022). We set LoRA rank as 16, fix learning rate as $10^{-5}$ with cosine schedule, and train models for 2 epochs on a single A100 GPU.

**Overall Results.** The rightmost column in Table 3 presents averaged results over all the samples (not simply over each row). (1) *Zero-shot*: Commercial LLMs consistently outperform open-source LLMs, and medium-sized (8B) open-source models outperform small (1B) ones. (2) After *instruction tuning*: All open-source models improve substantially; notably, Gemma3-1B (69.70) surpasses Gemini-2.5-Flash (65.08). These results indicate that instruction tuning can markedly enhance open-source models, narrowing the performance gap with and even outperform commercial LLMs.

**Task-Level Results.** (1) *Conventional Tasks.* In zero-shot settings, both commercial and open-source LLMs perform poorly on anomaly detection and classification, but open-source models improve markedly after instruction tuning (e.g., LLaMA-3.1-8B reaches 91.02 and 91.27). (2) *Advanced Tasks.* For characterization and comparison, commercial models outperform medium-sized open-source models, likely due to broader pretraining exposure. Data transformation and temporal relationship, especially PZ questions, remain difficult for all models. Instruction tuning boosts open-source performance, but there are still considerable room to improve. For example, for comparison task, best performing LLaMA-3.1-8B with instruction tuning only achieves 86.72 and 79.31.

**Question Type-Level Results.** Across the three question types (TF, MC, PZ), open-source models perform best on TF, worse on MC, and poorest on PZ. Performance on PZ is substantially lower than on TF and MC, in both zero-shot and tuned settings. Considerable room for improvement remains, e.g., the best PZ score is only 67.68 (LLaMA-3.1-8B after instruction tuning).

## 4.2 ANALYSIS

We use best performing commercial LLMs, i.e., Geimini-2.5-Flash and GPT-4.1, and open-source LLMs, i.e., LLaMA3.1-8B and Qwen3-8B to conduct further analysis. To examine their reasoning ability on the proposed TRQA Benchmark, we cover two key perspectives: *Accuracy Correlate Analysis* and *Task-Specific Analysis*.

### 4.2.1 ACCURACY CORRELATE ANALYSIS

**Input Lengths.** Figures 3 illustrate the relationship between input length and the number of correct predictions. The input length is calculated as *len(ts + description + domain + dataset + task + question_type + question)* with *String* type. Across all six models and five tasks, excluding the Temporal Relationship task, we observe a consistent trend that performance declines as input length increases, indicating that longer inputs correspond to more difficult questions. However, the Temporal Relationship task exhibits the opposite behavior, where accuracy improves with increasing input length. The analysis is shown in Figure 4, which the newly proposed PZ type question exhibits the opposite trend. The fact that PZ performance scales positively with length proves that the model is actively utilizing global context, all time series segments, to deduce the correct chronological order. This confirms the model is engaging in deductive reasoning rather than local pattern matching, proving that PZ type question is a rigorous probe for *Global Causal Reasoning*. More details are provided in Appendix E.1.

### 4.2.2 TASK SPECIFIC ANALYSIS

**Comparison.** We analyze model performance on the Comparison task, specifically investigating whether providing explicit domain-level context affects model accuracy. The task requires comparing two input time series, which we test under two conditions: (1) when both series originate from the same domain and (2) when they are from different domains. In both scenarios, the corresponding domain names are provided to the model as textual description. As shown in Table 8, we observe no significant performance difference between the same-domain and different-domain settings across either MC or TF questions. This suggests that the Comparison Task is domain invariant. More details are provided in Appendix E.2.

**Data Transformation.** We analyze model performance on the Data Transformation task, which is designed to evaluate a model's understanding of three transformation operators: Fourier Transform (FT), Wavelet Transform (WT), and First-Order Differencing (FOD). For each operator, we assess performance by measuring the accuracy on both MC and TF question formats. As shown in Table 9, for zero-shot evaluation, our key finding highlights a limitation in which both commercial and open-source models fail to provide accurate answers, except of FOD. More details are provided in Appendix E.2.

**Temporal Relationship.** Beyond the input length analysis in Section 4.2.1 demonstrating that PZ questions require *Global Casual Reasoning*, we further examined how domain-level information influences model performance on PZ questions. The results are summarized in Table 10, which Web and Sales domains remain the most challenging across both zero-shot and instruction-tuning settings. More details are provided in Appendix E.2.

## 4.3 HUMAN EVALUATION

We conduct human evaluations of the multi-LLM consensus labels for characterization and comparison (Section 3.3). Six Ph.D.-level experts mannually annotate 600 questions (300 each), serving as ground truth. Uncertain or problematic QA pairs are flagged, multiple answers allowed when valid, and explanations provided for disagreements with the benchmark.

Our evaluation yields two main findings. (1) Question quality: Uncertainty rates are low (5% for characterization, 7% for comparison), showing that most questions are clear. (2) Answer accuracy: For unambiguous cases, benchmark answers align with human judgments in 91.2% of characterization and 87.4% of comparison. These results indicate that the automatic pipeline produces reliable QA pairs, though comparison remains harder, with lower agreement and higher uncertainty (Figure 5). More details are provided in Appendix F.

## 5 CONCLUSION

TRQA establishes a large-scale comprehensive benchmark for time series question answering with 210k samples curated from 13 domains, covering 6 tasks and 3 types of questions, extending evaluation beyond traditional tasks, i.e., anomaly detection and classification, to advanced reasoning tasks, i.e., characterization, comparison, data transformation, temporal relationship reasoning. By spanning diverse domains, tasks, and question types, it offers a unified platform to probe the strengths and limitations of both commercial and open-source LLMs. Our results highlight that, despite progress with instruction tuning, substantial challenges remain—particularly for advanced reasoning and puzzling questions—underscoring the need for further research into models capable of deeper time series understanding.

## 6 REPRODUCIBILITY STATEMENT

We provide detailed descriptions of the TRQA benchmark to ensure reproducibility. The formulation of tasks and question types is introduced in 3.1, with data sources and preprocessing procedures explained in 3.2 and Appendix B. Benchmark construction, including sampling strategy, data partitioning, and design considerations, is described in 3.3 and Appendices C– D, while dataset statistics are summarized in 3.4. Evaluation protocols for different question types (TF, MC, PZ) are given in 3.5. The experimental setup, including model configurations, hyperparameters, and instruction-tuning details, is provided in 4. Human evaluation methodology and results are discussed in 4.3 and F. All data and code required to reproduce our results are available at the anonymous repository: `https://anonymous.4open.science/r/TRQA_benchmark-6737`.

## 7 ETHICS STATEMENT

This research was conducted in accordance with the ICLR Code of Ethics. The study did not involve human participants or animal subjects, and all datasets employed are publicly accessible and were utilized in strict compliance with their respective licensing agreements and data usage policies. The data contains no personally identifiable information (PII), and our experimental design inherently mitigates privacy and security risks.

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

## A THE USE OF LARGE LANGUAGE MODELS

We leverage Large Language Models (LLMs) from two perspectives: (1) Polishing the writing, where LLMs are used to refine the clarity, fluency, and consistency of the paper; and (2) Labeling, where LLMs assist in generating high-quality question-answer (QA) pairs and providing preliminary annotations, which are then validated or aggregated through consensus to create reliable ground-truth labels.

## B DATA COLLECTION

In this section, we detail the data sources, including *core datasets* (Appendix B.1), *anomaly detection datasets* (Appendix B.2), and *classification datasets* (Appendix B.3).

### B.1 CORE DATASETS

We extract data from multiple time-series datasets including: Australian Electricity Demand (Godahewa et al., 2021), BDG-2 Rat (Miller et al., 2020), GEF12 (Hong et al., 2014), ExchangeRate(Lai et al., 2018), FRED MD(McCracken & Ng, 2016), BIDMC32HR (Tan et al., 2020), PigArtPressure (Dau et al., 2019a), USBirths (Godahewa et al., 2021), Sunspot (Godahewa et al., 2021), Saugeenday (Godahewa et al., 2021), SubseasonalPrecip (Mouatadid et al., 2024), HierarchicalSales (Mancuso et al., 2021), M5 (Makridakis et al., 2022), PedestrianCounts (City of Melbourne, 2017), PEMS03 (Caltrans, 2025), UberTLCHourly (FiveThirtyEight, 2015),WikiDaily100k (Ansari et al., 2024b). Below are some more detailed descriptions on those dataets.

**Australian Electricity Demand.** A single long time series from the Monash Time Series Archive representing half-hourly electricity demand for Victoria, Australia in 2014 (17,520 observations), extracted from the R package `fpp2` (dataset name: "elecdemand"). Temperatures corresponding to each demand value are available in the original dataset.

**BDG-2 Rat .** From The Building Data Genome Project 2 (MIT License), consisting of measurements from 3,053 meters across 1,636 commercial buildings over 2016–2017. One or more meters per building measured total electrical, heating and cooling water, steam, solar energy, water, and irrigation usage. We use the whole-building electricity meter measurements from the Bear, Fox, Panther, and Rat sites, totaling 611 buildings (from the CSV file `electricity_cleaned.csv`).

**GEF12.** A benchmark compiled from the Global Energy Forecasting Competition 2012 (load forecasting tracks), containing 20 aggregated-level hourly load series and 11 temperature series from 2004-01-01 00:00 to 2008-06-30 05:00. Because the one-to-one correspondence between temperature and load series is not clearly defined, a common strategy is to use a single temperature series for all loads (here, the second temperature series). The dataset is competition-grade and was used without additional preprocessing; visualizations show obvious periodicity and seasonality in the aggregated loads.

**ExchangeRate.** Daily exchange rates for currencies of eight countries—Australia, United Kingdom, Canada, Switzerland, China, Japan, New Zealand, and Singapore—covering 1990 to 2016.

**FRED-MD.** 107 monthly time series of macro-economic indicators from the Federal Reserve Bank, starting from 1959-01-01, extracted from the FRED-MD database.

**BIDMC32HR.** Derived from BIDMC ICU recordings: PPG and respiratory signals/IP (sampling rate 125 Hz) from 53 adult patients, with breath annotations used to form reference targets in the source dataset. Following the adaptation in subsequent work, PPG and ECG are converted into 32-second sliding-window time series; the average heart rate (HR) in each 32 s window is the target. The datasets are split by randomly selecting 30% as test, yielding 5,550 training and 2,399 test time series.

**PigArtPressure.** Based on a source dataset from 52 pigs with three vital signs monitored before and after an induced injury. Three datasets are created: AirwayPressure (airway pressure), ArtPressure (arterial blood pressure), and CVP (central venous pressure).

**US Births.** A single long daily time series of the number of births in the United States from 1969-01-01 to 1988-12-31 (7,305 observations), extracted from the R package `mosaicData`.

**Sunspot.** A single long daily time series of sunspot numbers from 1818-01-01 onward, with additional related series (monthly means, smoothed series, yearly totals, hemispheric series) in the original source. The repository used here contains the daily series from 1818-08-01 to 2020-05-31 and includes both the raw data (with missing values) and an LOCF-imputed version.

**Saugeen.** A single long daily time series of the Saugeen River mean flow at Walkerton (in cubic meters per second) from 1915-01-01 to 1979-12-31 (23,741 observations), extracted from the R package `deseasonalize` (dataset name: "SaugeenDay").

**Subseasonal Precipitation.** Extracted from SubseasonalClimateUSA: daily precipitation measurements (millimeters) for a single $1.5° \times 1.5°$ latitude–longitude grid cell, covering 1948–1978.

**Hierarchical Sales.** 118 daily time series of SKU-level sales for four national pasta brands from 2014-01-01 to 2018-12-31, including a binary indicator for promotion. The series can be organized into a three-level hierarchy.

**M5.** The M5 "Accuracy" competition dataset requiring point forecasts for 30,490 bottom-level daily series that aggregate to 42,840 time series representing hierarchical unit sales for Walmart. The competition paper details the implementation, results, top methods, and implications for forecasting research.

**Pedestrian Counts.** Hourly pedestrian counts from 66 sensors in Melbourne starting from May 2009. The original data are updated monthly; the repository snapshot used here contains counts up to 2020-04-30.

**PEMS03.** Datasets sourced from Caltrans PeMS, which collects 30-second traffic readings and aggregates them into 5-minute intervals (288 time steps per day). Road network structure is derived from connectivity status and actual distances between sensors.

**Uber TLC Daily.** Counts of Uber pick-ups from various New York City locations between January and June 2015, obtained from FiveThirtyEight's "uber-tlc-foil-response" repository and aggregated at hourly and daily resolutions.

**WikiDaily10k.** Daily traffic data for 10,000 Wikipedia pages.

Table 4: Summary of the core datasets.

| dataset_name | total_data_point | domain |
| --- | --- | --- |
| AustralianElectricityDemand | 1,153,584 | energy |
| BDG-2 Rat | 4,728,288 | energy |
| GEF12 | 788,280 | energy |
| ExchangeRate | 56,096 | finance |
| FRED MD | 76,612 | finance |
| BIDMC32HR | 8,000,000 | healthcare |
| PigArtPressure | 624,000 | healthcare |
| USBirths | 7,275 | healthcare |
| Sunspot | 73,924 | nature |
| Saugeenday | 23,711 | nature |
| SubseasonalPrecip | 9,760,426 | nature |
| HierarchicalSales | 212,164 | sales |
| m5 | 58,327,370 | sales |
| PedestrianCounts | 3,130,762 | transport |
| PEMS03 | 9,382,464 | transport |
| UberTLCHourly | 1,129,444 | transport |
| WikiDaily100k | 274,099,872 | web |

## B.2 ANOMALY DETECTION DATASET

We extract data from multiple time-series anomaly detection benchmarks Paparrizos et al. (2022); Su et al. (2019), including ECG (Moody & Mark, 2001), SMD (Su et al., 2019), MGAB (Thill et al.) Genesis (von Birgelen & Niggemann, 2018), GHL (Filonov et al., 2016), Occupancy (Candanedo &

Feldheim, 2016). These datasets span various domains, including healthcare (ECG), mathematical biology (MGAB), spacecraft telemetry (Genesis), industrial control system (GHL), environmental sensing (Occupancy), cyber-security on IT Operations (SMD). The statistics of these datasets are shown in Table 5. To address class imbalance, we count the number of anomalous sequences and randomly select an equal number of normal sequences, resulting in a balanced dataset. Below are the meta information for each dataset.

**MGAB.** This dataset is composed of Mackey-Glass time series with non-trivial anomalies. Mackey-Glass time series exhibit chaotic behavior that is difficult for the human eye to distinguish.

**ECG.** This dataset is a standard electrocardiogram dataset and the anomalies represent ventricular premature contractions. The ECG recordings were made using Del Mar Avionics model 445 two-channel reel-to-reel Holter recorders, and the analog signals were recreated for digitization using a Del Mar Avionics model 660 playback unit. The digitization rate (360 samples per second per channel) was chosen to accommodate the use of simple digital notch filters to remove 60 Hz (mains frequency) interference.

**Genesis.** This dataset is a portable pick-and-place demonstrator which uses an air tank to supply all the gripping and storage units. Data samples were taken through an OPC connection with a resolution of 50 milliseconds for a total of 42 production cycles. The first 38 production cycles contain only normal behavior and were used to train the selforganizing map for both experiments shown in this section. Two of the 4 remaining cycles contain anomalous behavior and are used for the anomaly detection.

**GHL.** This dataset is a Gasoil Heating Loop Dataset and contains the status of 3 reservoirs such as the temperature and level. Anomalies indicate changes in max temperature or pump frequency. Type of cyber attack to the normal process logic is the unauthorized change of max Receiving Tank level. By changing the time of attack and the value of the hacked max Receiving Tank level, we generated many anomalous data sets used for fault detection.

**Occupancy.** This dataset contains experimental data of room occupancy, such as temperature, humidity, light, and CO2. Ground-truth occupancy was obtained from time stamped pictures that were taken every minute.

**SMD.** SMD (Server Machine Dataset) is collected from a large Internet company. The data is sampled every 5 seconds. Labels denote whether a point is an anomaly and the dimensions contribute to every anomaly.

Table 5: Summary of anomaly detection datasets.

| Name | # Samples | Domain |
|------|-----------|--------|
| ECG | 17,862 | Healthcare |
| SMD | 58,888 | Cyber-security on IT Operations |
| MGAB | 376 | mathematical biology |
| Genesis | 274 | Spacecraft Telemetry |
| GHL | 768 | Industrial Control System |
| Occupancy | 8,178 | Environmental Sensing |

### B.3 CLASSIFICATION DATASET

We extract data from the UCR Archive (Dau et al., 2019a). To create a focused subset for our study, we applied two primary selection criteria: we included only datasets with four or fewer classes and time series with a sequence length of 400 time points or less. Through our selection, we extract data from 37 benchmarks in the UCR Archive, including SonyAIBORobotSurface1 & SonyAIBORobotSurface2 (Mueen et al., 2011), FreezerRegularTrain & FreezerSmallTrain (Murray, 2015), ToeSegmentation1 & ToeSegmentation2 (Ye & Keogh, 2011), TwoPatterns (Geurts, May 2002), CBF (Saito & Coifman, 1994), Wafer & ECG200 (Olszewski et al., 2001), TwoLeadECG, ECGFiveDays, DistalPhalanxOutlineCorrect & MiddlePhalanxOutlineCorrect & ProximalPhalanxOutlineCorrect & DistalPhalanxOutlineAgeGroup & MiddlePhalanxOutlineAgeGroup & ProximalPhalanxOutlineAgeGroup & PhalangesOutlinesCorrect (Bagnall & Davis, 2014), MoteStrain (Sun

et al., 2005), GunPointMaleVersusFemale & GunPointOldVersusYoung & GunPointAgeSpan & GunPoint (Ratanamahatana & Keogh, 2005), Strawberry (Holland et al., 1998), ItalyPowerDemand (Keogh et al., 2006), Chinatown, BME, PowerCons, DodgersLoopWeekend & DodgersLoopGame (Ihler et al., 2006), DiatomSizeReduction, SmoothSubspace (Huang et al., 2016), UMD, Wine, Coffee (Briandet et al., 1996), and ArrowHead (Ye & Keogh, 2009). These datasets span various domains, including robotics, energy, healthcare, synthetic, manufacturing, nature, and transport. The statistics of these datasets are shown in Table 6.

Table 6: Classification data used in our experiments.

| Name | # Samples | # Classes | Domain |
|------|-----------|-----------|--------|
| SonyAIBORobotSurface1 | 486 | 2 | Robotics |
| SonyAIBORobotSurface2 | 771 | 2 | Robotics |
| FreezerRegularTrain | 2,404 | 2 | Energy |
| FreezerSmallTrain | 2,353 | 2 | Energy |
| ToeSegmentation1 | 210 | 2 | Healthcare |
| ToeSegmentation2 | 129 | 2 | Healthcare |
| TwoPatterns | 3,999 | 4 | Synthetic |
| CBF | 757 | 3 | Synthetic |
| Wafer | 5,744 | 2 | Manufacturing |
| ECG200 | 159 | 2 | Healthcare |
| TwoLeadECG | 923 | 2 | Healthcare |
| ECGFiveDays | 704 | 2 | Healthcare |
| DistalPhalanxOutlineCorrect | 690 | 2 | Healthcare |
| MiddlePhalanxOutlineCorrect | 731 | 2 | Healthcare |
| ProximalPhalanxOutlineCorrect | 688 | 2 | Healthcare |
| DistalPhalanxOutlineAgeGroup | 423 | 3 | Healthcare |
| MiddlePhalanxOutlineAgeGroup | 435 | 3 | Healthcare |
| ProximalPhalanxOutlineAgeGroup | 485 | 3 | Healthcare |
| PhalangesOutlinesCorrect | 2,076 | 2 | Healthcare |
| MoteStrain | 1,012 | 2 | Nature |
| GunPointMaleVersusFemale | 362 | 2 | Healthcare |
| GunPointOldVersusYoung | 356 | 2 | Healthcare |
| GunPointAgeSpan | 368 | 2 | Healthcare |
| GunPoint | 169 | 2 | Healthcare |
| Strawberry | 786 | 2 | Nature |
| ItalyPowerDemand | 890 | 2 | Energy |
| Chinatown | 293 | 2 | Transport |
| BME | 137 | 3 | Synthetic |
| PowerCons | 294 | 2 | Energy |
| DodgersLoopWeekend | 111 | 2 | Transport |
| DodgersLoopGame | 115 | 2 | Transport |
| DiatomSizeReduction | 248 | 4 | Nature |
| SmoothSubspace | 236 | 3 | Synthetic |
| UMD | 148 | 3 | Synthetic |
| Wine | 85 | 2 | Nature |
| Coffee | 48 | 2 | Nature |
| ArrowHead | 175 | 3 | Nature |

---

**Algorithm 1:** Hierarchical Random Sampling

---

**Input:** Domains $\mathcal{M}$;
   Datasets $\mathcal{D}(m)$ for each domain $m \in \mathcal{M}$;
   Sequences $\mathcal{S}(d)$ for each dataset $d \in \mathcal{D}$;
   Segment length $l$
**Output:** Segment $s_{t:t+l-1}$

---

$m \leftarrow \text{UniformPick}(\mathcal{M})$;                       `// Randomly select a domain`
$d \leftarrow \text{UniformPick}(\mathcal{D}(m))$;    `// Randomly select a dataset in the domain`
$s \leftarrow \text{UniformPick}(\mathcal{S}(d))$;    `// Randomly select a seq.  from the dataset`
$t \leftarrow \text{UniformPick}\{1, \ldots, |s| - l + 1\}$;    `// Randomly select a start index`
**return** $s_{t:t+l-1}$;                             `// Return the segment`

---

## C   Benchmark Construction

In this section, we provide extra content about the construction process for each task and provide examples of each task.

### C.1   Hierarchical Uniform Sampling

For all the advanced reasoning tasks, including characterization, comparison, data transformation and temporal relationship, all the input time series are sampled from the *core dataset* (Appendix B.1). To ensure a balanced distribution over domains, datasets and sequences, we use *Hierarchical Uniform Sampling* presented in Algorithm 1 to obtain samples.

### C.2   Data Bias

Unless otherwise specified, all samples have a random length in $[32, 256]$, and are z-scored to reduce data bias. The term data bias refers specifically to scale-based shortcuts or magnitude variance across heterogeneous domains, rather than semantic or sampling bias. We justify the use of z-score normalization on 2 main grounds: (1) *Preventing Magnitude-Based Shortcuts*, (2) *Standard Practice and Task Alignment*.

*Preventing Magnitude-Based Shortcuts*: TRQA is a unified benchmark that aggregates data from 13 distinct domains, each possessed of vastly different magnitudes and units. Without normalization, large language models (LLMs) could exploit these scale differences as shortcuts to identify the source domain or dataset without performing genuine temporal reasoning. Normalization prevents this risk, forcing the model to rely on structural reasoning rather than memorizing absolute value ranges.

*Standard Practice and Task Alignment*: While real-world data is indeed not standardized, normalization is a ubiquitous and necessary preprocessing step in the time series literature to ensure numerical stability and cross-domain comparability. This approach aligns with established protocols in widely used benchmarks such as the UCR Archive (Dau et al., 2019b), and recent time series foundation model studies like Time-LLM (Jin et al., 2024) and Chronos (Ansari et al., 2024a), which consistently utilize normalization or scaling to handle distribution shifts. Additionally, the core objective of TRQA is to evaluate reasoning capabilities. Z-score normalization is a linear transformation that preserves the fundamental properties required for these tasks while removing the confounding factor of arbitrary absolute magnitudes.

### C.3   Characterization

The characterization task assesses the model's capability to reason about fundamental properties of time series, including trend, seasonality, and dispersion. Questions are posed as TF or MC, and final answers are determined through multi-LLM consensus.

Table 7: Topics and Sub-Topics for Time Series Analysis

| Topic | Sub-Topics |
|---|---|
| Trend | trend directions, trend types, trend shapes, trend strength, structural breaks, global and local trends |
| Seasonality | seasonality period, seasonal strength, multiple seasonality patterns, changing seasonality |
| Cyclicity | amplitude, peaks and trough, duration |
| Noise | noise level, global and local noise |
| Stationarity | stationarity strength, global and local stationarity, types of non-stationarity |
| Autocorrelation | types of autocorrelation, autocorrelation structures, lags, mean-reversion, persistence of autocorrelation |
| Dispersion | basic measures of variability (variance level), relative measures (signal-to-noise ratio level), coefficient of variation level, time-varying dispersion (volatility, heteroskedasticity), entropy, multi-scale dispersion |
| Shape | global shapes, local shapes, shapelets, motifs, curves, change points, pattern complexity |
| Irregularity | mean shift, variance shift, trend shift, seasonality irregularity, cyclic shift, distributional change, structural breaks, autocorrelation change |
| Correlation (Comparison only) | causal relationship, correlation strength, correlation types, correlation direction, cross-correlation, time-varying correlation (rolling correlation), lagged correlation, global and local correlations, correlation of decomposed components |

Each instance consists of a univariate time series sample $\mathbf{x}$ with associated metadata (text description, domain, dataset). Given a sample $\mathbf{x}$ and its metadata, we instruct GPT-4o (Hurst et al., 2024) to generate one QA pair per instance using a randomly selected subset of one to three topics (from Table 7) and a question type (TF or MC). The process is as follows.

*Step 1: Captioning & sub-topic selection.* GPT first produces a short, neutral caption summarizing visible patterns (e.g., "gradual upward drift with weak weekly oscillation"). For each chosen topic, a sub-topic is sampled uniformly at random, e.g., trend, seasonality and dispersion.

*Step 2: QA synthesis.* GPT generates a TF or MC question grounded in $\mathbf{x}$, the caption, and the selected sub-topics.

*Step 3: Self-verification.* GPT performs a self-check and outputs a confidence score in [0,1]. We retain QA pairs only if confidence $\geq 0.95$.

*Step 4: Multi-LLM consensus.* We query GPT-4.1, Gemini-2.5-Flash, and Claude-3.5-Sonnet using the same prompt, which includes the generated question along with its allowed answer choices (for both TF and MC formats), and collect their responses. To determine the final label, we adopt a weighted majority voting scheme among these three models and GPT-4o's original answer. Specifically, GPT-4.1 and Gemini-2.5-Flash are assigned higher weights of 1.5 each, reflecting their superior performance in preliminary evaluations, while Claude-3.5-Sonnet and GPT-4o are each assigned a weight of 1.0. The option with the highest total weighted vote is selected as the consensus answer. If a tie occurs—i.e., two or more answers receive the same highest weighted score—the corresponding QA pair is discarded to avoid introducing ambiguity or noise into the dataset. This ensemble-based strategy mitigates single-model biases, smooths out random errors, and produces more reliable and stable labels, which are crucial for ensuring the benchmark's quality.

Here's the *system* prompt template.

```
You are an expert of time series analysis.
```

```
1. Generate a meta_caption solely based on the meta information within 50
     words.
2. Generate a detailed_caption based on both meta information and time
     series within 100 words.
3. Generate a {} based on the time series, meta_caption, detailed_caption
     and the more detailed question instructions.
4. Generate a correct answer {} for your question.
5. A successful generation must meet the following conditions:
(1) there is only one correct answer;
(2) the question stricktly follows the instructions;
(3) the answer of the question cannot be easily derived from the
     meta_caption;
(4) the question should be about the time series itself without invovling
     external knowledge;
(5) do not repeat the input time series in questions or answers.
6. Show your confidence of your determination of success within 0-1.
```

Here's the *user* prompt template.

```
The time series is {}.
Its meta information is {}.
The question must be about all these topics: {}.
The sub-topics of {} includes but not limited to {}.
First think about the all possible sub-topics and their taxonomy.
Then randomly pick a sub-topic from each topic ({}) to generate the
     question and answer pairs.
```

## C.4 COMPARISON

The comparison task assesses the model's ability to reason about the relative characteristics of two time series, such as overall shape, temporal alignment, and correlation patterns. Similar to the characterization task, this task is also formulated as either TF or MC questions, where the model must identify similarities or differences between the given pair of sequences. The characteristics evaluated in the task are directly drawn from the standardized taxonomy of Topics and Sub-topics (from Table 7), which is shared with the Characterization task.

To construct the comparison set, we first obtain an anchor sample $\mathbf{x}$ from a specific domain $M$, dataset $D$, and sequence $S$. Given this anchor $\mathbf{x}$, we generate a set of ten comparison samples $\mathbf{x}'1, \ldots, \mathbf{x}'10$, each having the same length as $\mathbf{x}$. These samples are drawn in a structured manner to represent varying degrees of similarity: one from the same sequence $S$, two from different sequences within the same dataset $D$, three from other datasets within the same domain $M$, and four from entirely different domains. This tiered sampling strategy creates a natural hierarchy of difficulty, challenging the model to distinguish between subtle intra-sequence similarities and broader cross-domain differences.

Finally, we apply a process similar to the characterization task to generate QA pairs, where GPT-based models produce questions and candidate answers. The questions are then refined and validated through multi-LLM consensus to ensure accuracy and reduce bias, resulting in high-quality, reliable evaluation data for this task.

## C.5 DATA TRANSFORMATION

The data transformation task evaluates the model's ability to infer and reason about the transformation relationship between an input time series and its transformed counterpart. These transformations are generated using well-established signal processing techniques, including the Fourier transform, wavelet transform, and first-order differencing, which are widely used in time series analysis to reveal underlying structures or remove trends. This task is particularly challenging because it requires the model to not only recognize the patterns in the raw input series but also to understand how specific mathematical operations alter these patterns.

We use predefined templates to formulate the task as either TF or MC questions. For TF questions, the model is asked to determine whether a given candidate sequence is indeed the correct transfor-

mation of the input time series $\mathbf{x}$ (e.g., whether it is the Fourier transform result of $\mathbf{x}$). For MC questions, the model must select the correct transformed sequence from multiple candidates, given both the input series $\mathbf{x}$ and the specified transformation operation (e.g., Fourier transform).

To ensure accuracy and consistency, all transformations are computed using professional and reliable scientific libraries (Harris et al., 2020; Virtanen et al., 2020). The correct transformation is generated directly from the input $\mathbf{x}$, while distractor sequences are created by applying the same transformation to randomly sampled, unrelated time series $\mathbf{x}'$. This setup forces the model to carefully analyze the relationship between the input and its transformation rather than relying on superficial similarities, providing a robust evaluation of its reasoning ability.

Here's the template to construct question.

```
The time series is {}.
Its meta information is {}.
The question must be about all these topics: {}.
The sub-topics of {} includes but not limited to {}.
First think about the all possible sub-topics and their taxonomy.
Then randomly pick a sub-topic from each topic ({}) to generate the
    question and answer pairs.
```

### C.6  TEMPORAL RELATIONSHIP

The Temporal Relationship task is a discriminative sequence-level reasoning task, rather than a generative forecasting task. The task evaluates a model's ability to infer and reason about the temporal structure among sequential patches of a time series. Specifically, the task evaluates whether a model can understand the structural continuity and chronological dependencies of time series patches, testing 3 core reasoning capabilities: *Structural Continuity*, *Chronological Reasoning*, and *Contextual Discrimination*. (1) *Structural Continuity* tests whether the model can identify which candidate segment shares the underlying temporal dynamics required to validly continue a given trajectory. (2) *Chronological Reasoning* tests whether the model can reconstruct the correct temporal order of shuffled patches. (3) *Contextual Discrimination* tests the model's ability to distinguish the true continuation from "plausible" but incorrect alternatives that may share similar global statistics but lack local continuity. This task is formulated as true-or-false (TF), multiple-choice (MC), or puzzling (PZ) questions.

Given the first chronological patch $\mathbf{x}$: (1) A TF question asks the model to determine whether a candidate patch $\mathbf{y}$ is the immediate successor of $\mathbf{x}$. (2) An MC question requires the model to select the correct next patch from four candidates $[\mathbf{y}_1, \mathbf{y}_2, \mathbf{y}_3, \mathbf{y}_4]$.

The false candidates in both TF and MC settings are randomly sampled from the full dataset but are guaranteed to come from sequences different from that of $\mathbf{x}$, preventing the model from simply memorizing patterns. For PZ questions, the model is presented with four shuffled successor patches of $\mathbf{x}$ and must reconstruct their correct chronological order, which poses a greater challenge as it requires deeper temporal reasoning. All questions are generated using predefined templates to ensure consistency and diversity.

We use the following question template to construct questions.

```
Which of the following choices is most likely the future continuation of
    the given time series?
Respond ONLY with the letter of the correct choice (A, B, C, or D)

Choices:
A: {}
B: {}
C: {}
D: {}
```

```
Is the following patch the future continuation of the given time series?
{}
Respond ONLY with the letter of the correct choice (T or F).
```

```
Choices:
T: True.
F: False.
```

## C.7 ANOMALY DETECTION

First, all time-series data are standardized using z-score normalization to remove scale effects across different features. Next, we randomly sample a subsequence of length $T$, where $T \in [32, 256]$, from each time-series instance to capture varying temporal dynamics. To address class imbalance, we count the number of anomalous sequences and randomly select an equal number of normal sequences, resulting in a balanced dataset. Finally, we enrich each sample with meta information, domain information, the normalized time-series subsequence, and its corresponding label.

Here's the question template.

```
Determine whether the given time series contains anomalies.
Respond ONLY with the letter of the correct choice (T or F).

Choices:
T: True.
F: False.
```

## C.8 CLASSIFICATION

Information about the time series and the task is given in the text description. Here's the template to construct questions.

```
Classify the given time series into one of the categories below.
Respond ONLY with the letter of the correct choice (A, B).

Choices:
A: {}
B: {}
```

## D EXAMPLES

In this section, we show some examples of the constructed QA pairs.

**TRQA — Sample 1**

Time Series Info

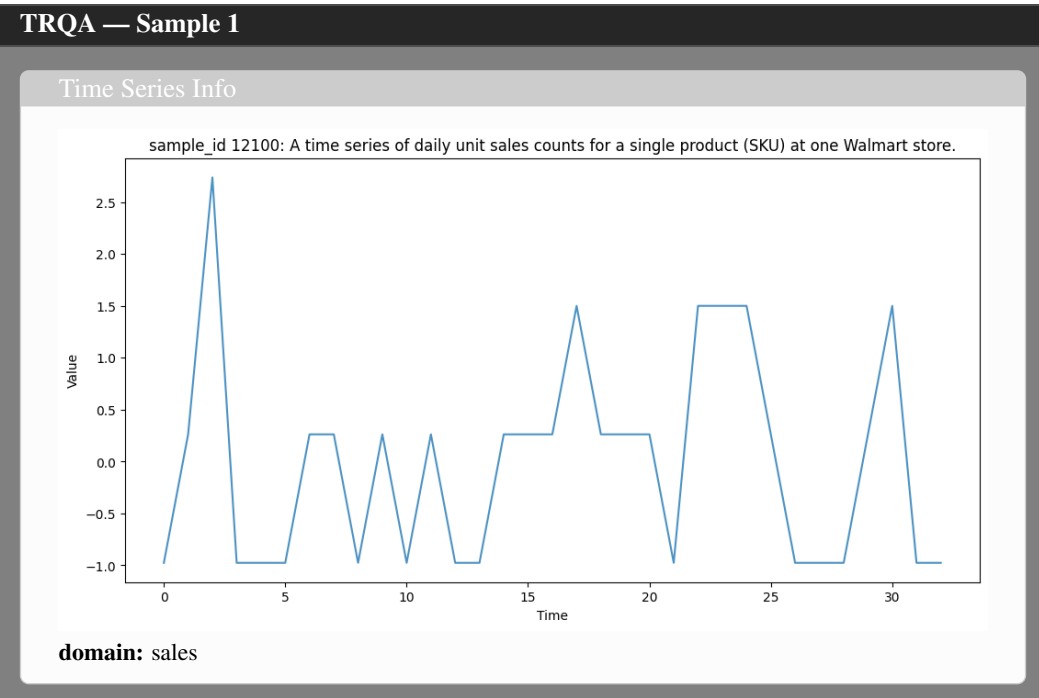

sample_id 12100: A time series of daily unit sales counts for a single product (SKU) at one Walmart store.

**domain:** sales

Question & Answer

**question:** Which statement best describes the overall characteristics of this time series with regard to its motif, variability, and trend? Respond *ONLY* with the letter of the correct choice (A, B, C, or D).
Choices:

A: The time series has a simple motif, low variability, and lacks a consistent upward or downward trend.

B: The time series features a complex motif, high variability, and an upward trend through the series.

C: The time series has a constant value, no motif, and displays a strong downward trend.

D: The time series contains random patterns with high variability and a significant upward trend.

**question_type:** multiple_choices    **task:** characterization

**answer:** A

## TRQA — Characterization Sample 1

### Time Series Info

**Description:** A numerical sequence of hourly aggregated Uber pickup counts (integer values) for a single New York City taxi zone, where each element represents the total number of pickups recorded in that zone during one hour.

**Question Type:** TF **Domain:** transport **Dataset:** uber_tlc_hourly

### Question & Answer

**Question:** Does the time series exhibit constant variance throughout, indicating no change in volatility, along with a clearly defined global shape? Respond ONLY with the letter of the correct choice (T or F).
Choices:

T: True.

F: False.

**Answer:** F

## TRQA — Characterization Sample 2

### Time Series Info

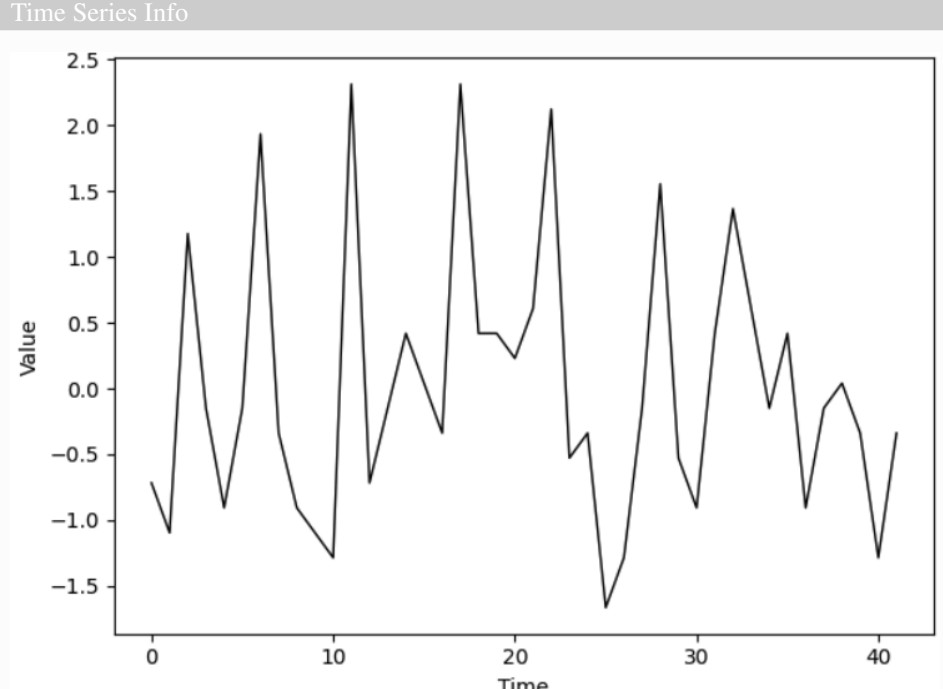

**Description:** A time series of daily page-view counts for a single English Wikipedia article.
**Question Type:** MC **Domain:** web **Dataset:** wiki_daily_100k

### Question & Answer

**Question:** Which of the following best describes a prominent motif and cyclic feature observed in the time series? Respond ONLY with the letter of the correct choice (A, B, C, or D).
Choices:

A: A recurring increase in values every few days with a noticeable peak at day 11.

B: A consistent downward trend over the entire period without any peaks.

C: An irregular pattern with no identical sequences or cycles.

D: A repeating cycle of gradual increase and sudden drop every ten days.

**Answer: A**

## TRQA — Comparison Sample 1

### Time Series Info

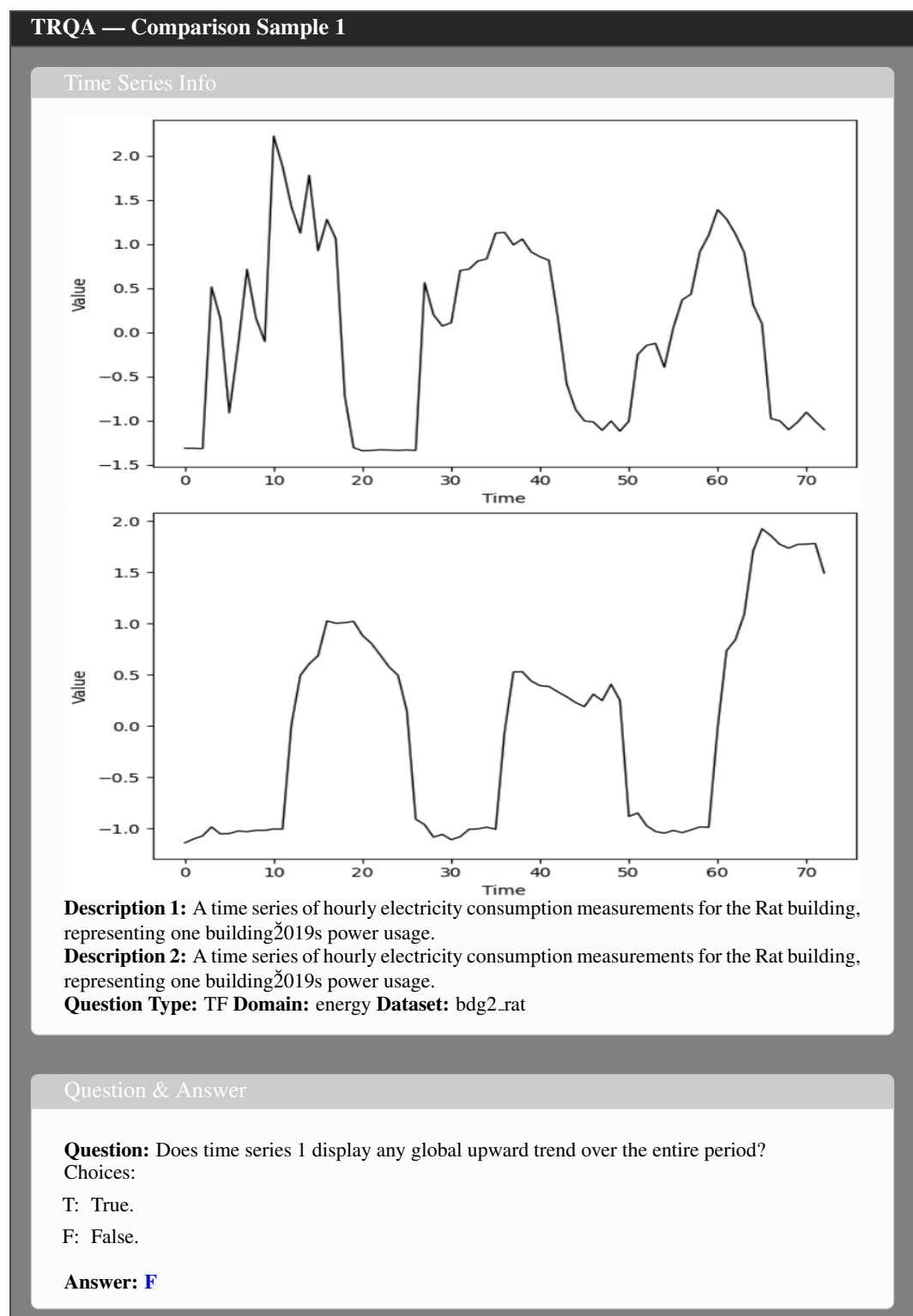

**Description 1:** A time series of hourly electricity consumption measurements for the Rat building, representing one buildingž2019s power usage.
**Description 2:** A time series of hourly electricity consumption measurements for the Rat building, representing one buildingž2019s power usage.
**Question Type:** TF **Domain:** energy **Dataset:** bdg2_rat

### Question & Answer

**Question:** Does time series 1 display any global upward trend over the entire period?
Choices:

T: True.

F: False.

**Answer:** F

**TRQA — Comparison Sample 2**

Time Series Info

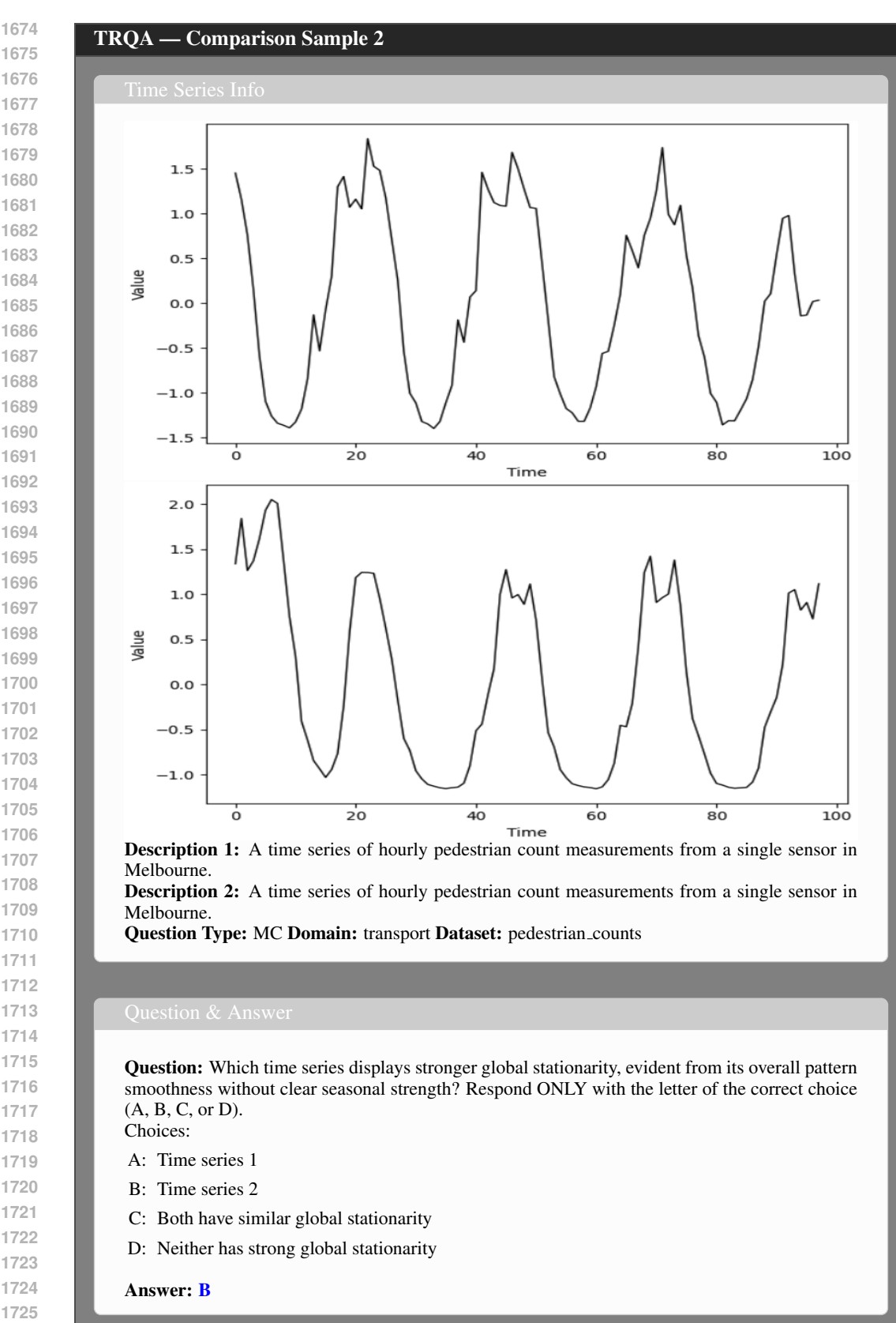

**Description 1:** A time series of hourly pedestrian count measurements from a single sensor in Melbourne.

**Description 2:** A time series of hourly pedestrian count measurements from a single sensor in Melbourne.

**Question Type:** MC **Domain:** transport **Dataset:** pedestrian_counts

Question & Answer

**Question:** Which time series displays stronger global stationarity, evident from its overall pattern smoothness without clear seasonal strength? Respond ONLY with the letter of the correct choice (A, B, C, or D).

Choices:

A: Time series 1

B: Time series 2

C: Both have similar global stationarity

D: Neither has strong global stationarity

**Answer: B**

**TRQA — Data Transformation Sample 1**

Time Series Info

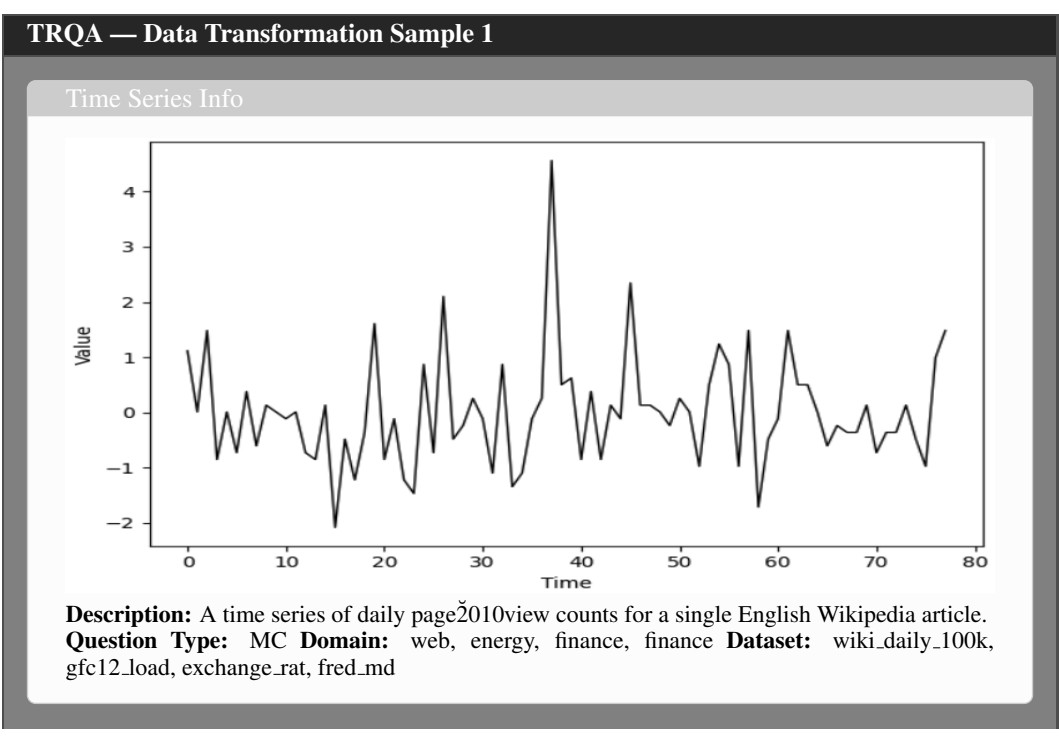

**Description:** A time series of daily pageˇ2010view counts for a single English Wikipedia article. **Question Type:** MC **Domain:** web, energy, finance, finance **Dataset:** wiki_daily_100k, gfc12_load, exchange_rat, fred_md

Question & Answer

**Question:** Which of the following choices is most likely the First Order Difference of the given time series? Respond ONLY with the letter of the correct choice (A, B, C, or D).
Choices:

A:

B:

C:

D:

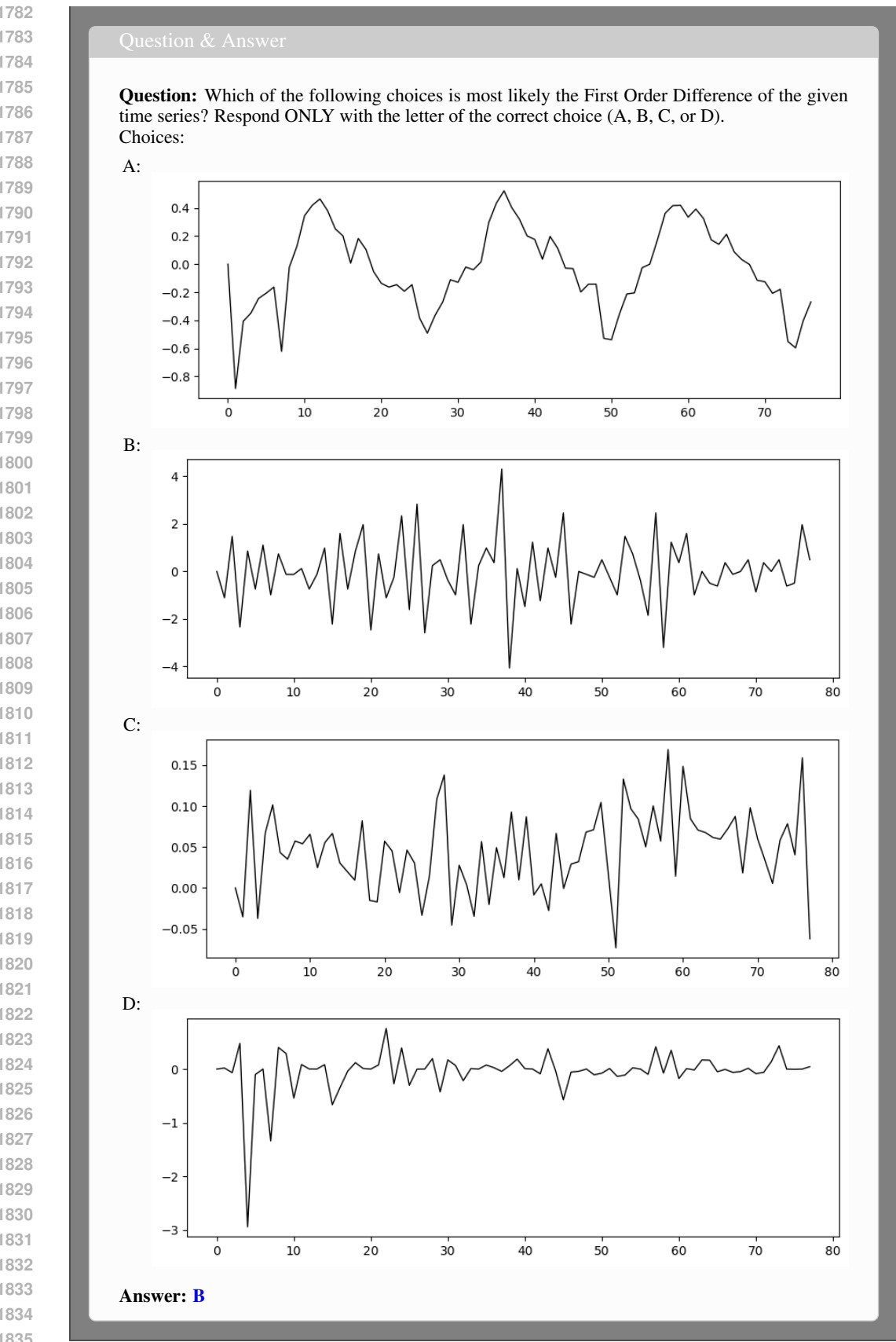

**Answer: B**

## TRQA — Data Transformation Sample 2

### Time Series Info

**Description:** A time series of daily relative sunspot number measurements, where each value represents the quantified count of sunspot activity on the Suň2019s visible disk.

**Question Type:** TF **Domain:** nature, energy **Dataset:** Nature_sunspot, gfc12_load

### Question & Answer

**Question:**
Is the following sequence the First Order Difference of the given time series?

Respond ONLY with the letter of the correct choice (T or F).
Choices:

T: True.

F: False.

**Answer: T**

**TRQA — Temporal Relationship Sample 1**

Time Series Info

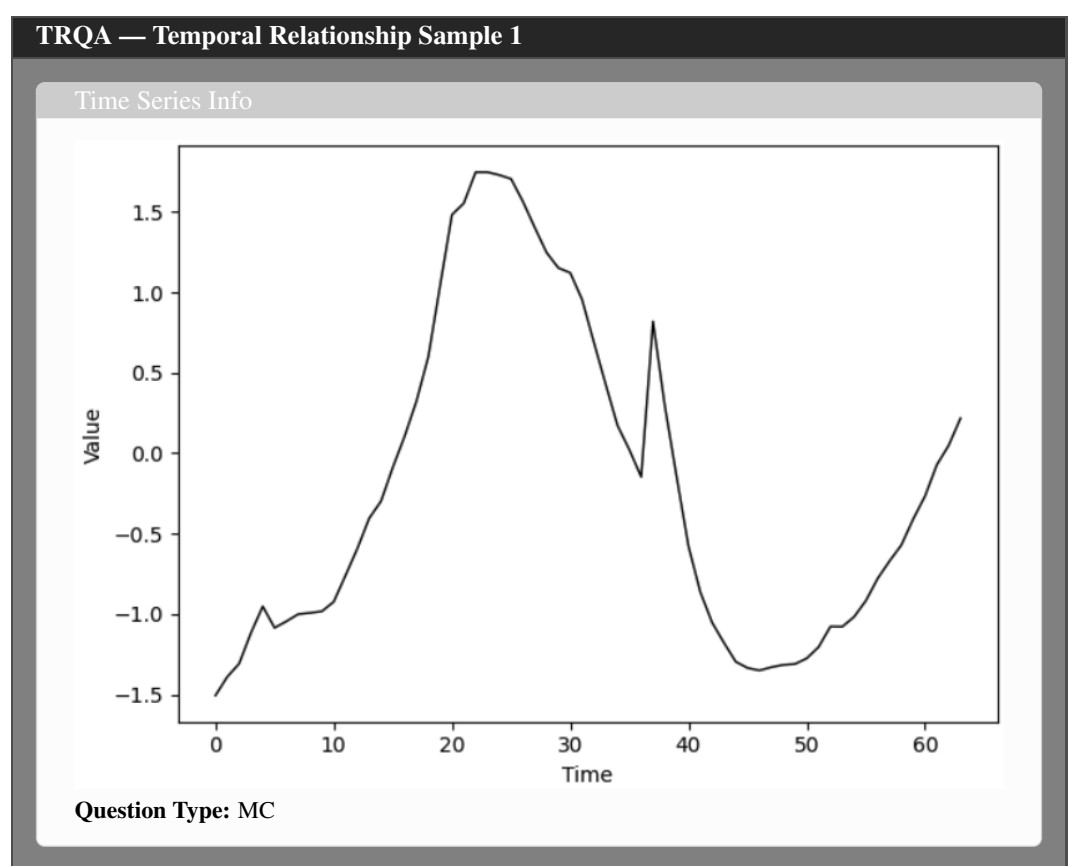

**Question Type:** MC

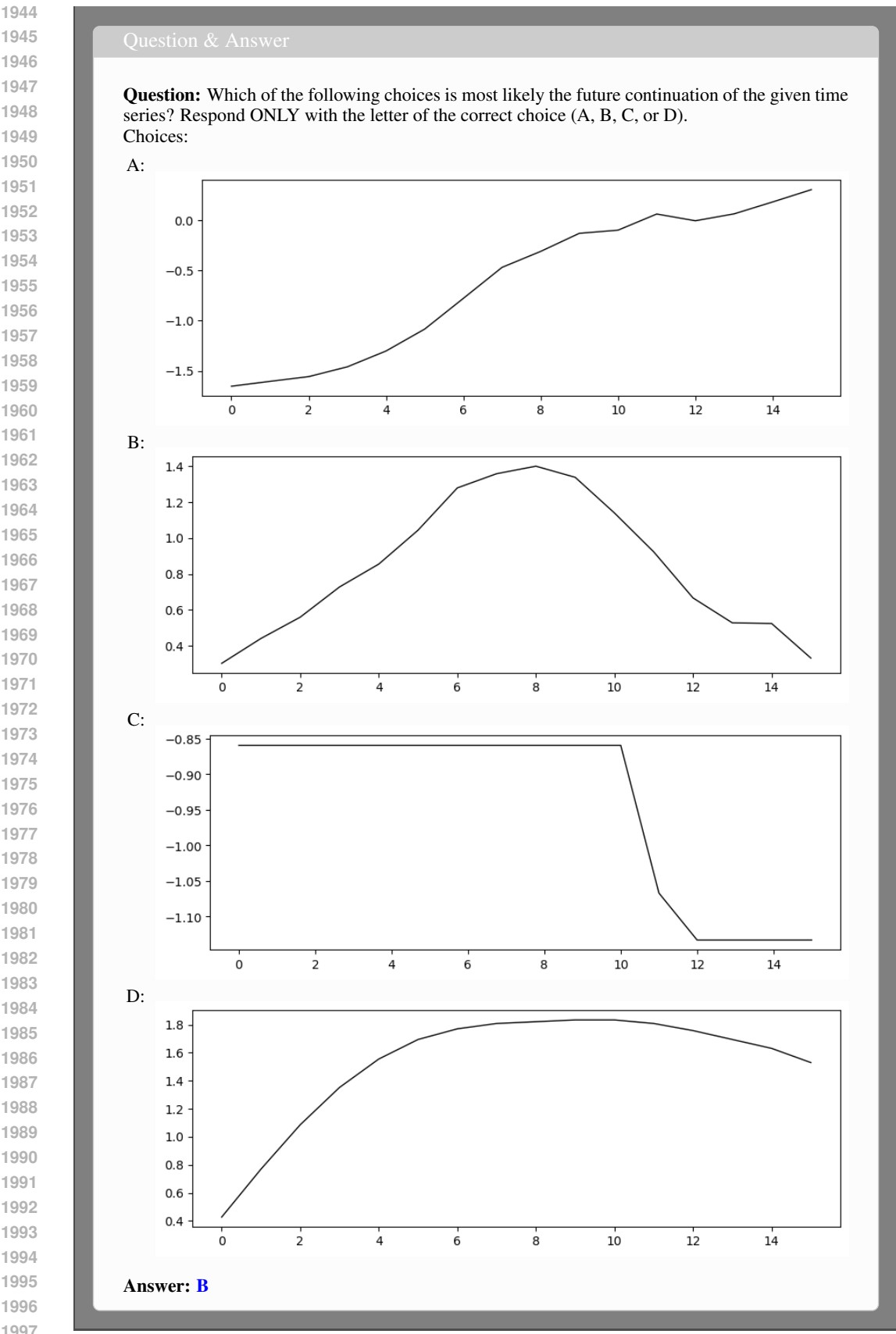

**TRQA — Temporal Relationship Sample 2**

Time Series Info

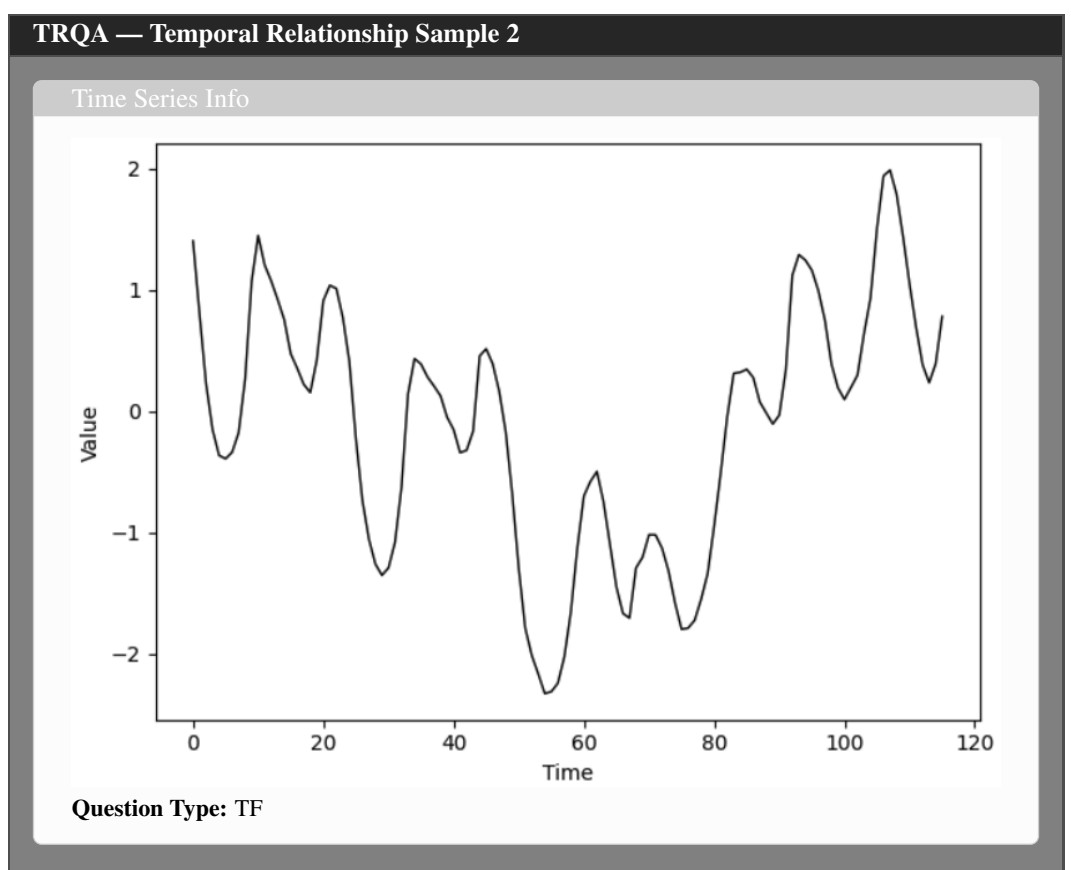

**Question Type:** TF

## Question & Answer

**Question:**
Is the following patch the future continuation of the given time series?

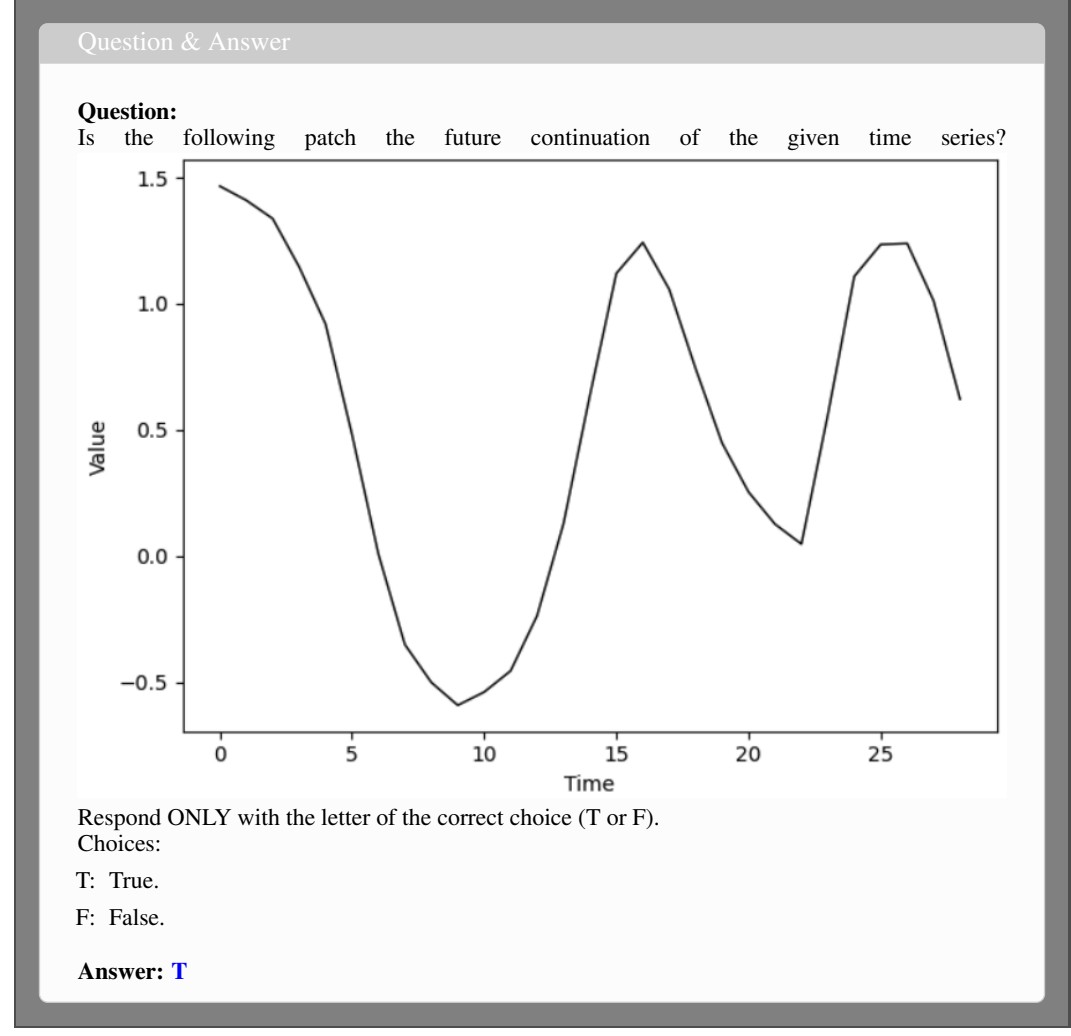

Respond ONLY with the letter of the correct choice (T or F).
Choices:

T: True.

F: False.

**Answer: T**

## TRQA — Temporal Relationship Sample 3

### Time Series Info

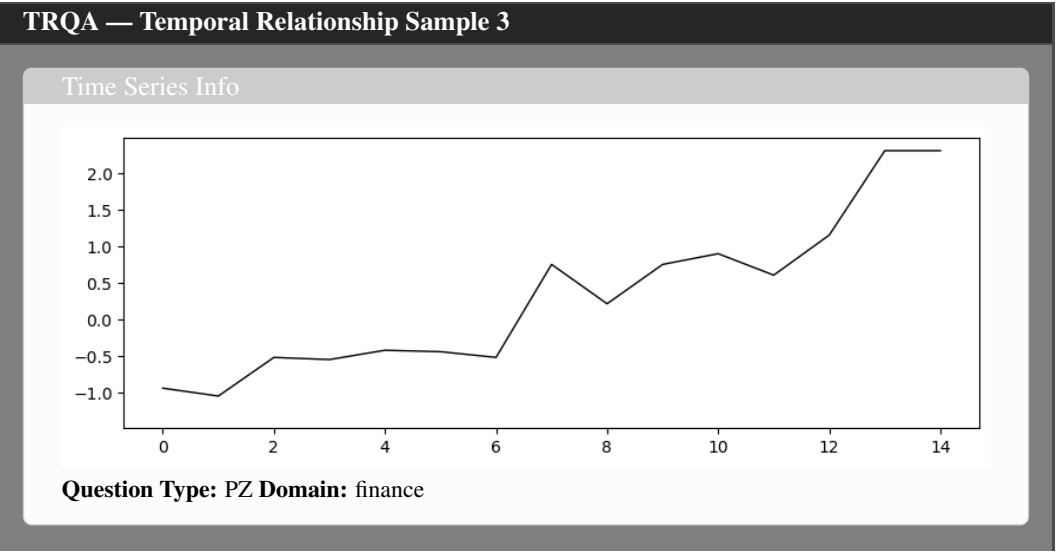

**Question Type:** PZ **Domain:** finance

**Question:** The given time series is the first patch of the sequence. Below are the remaining patches, labeled as A, B, C, and D. Arrange A, B, C, D in the correct order to reconstruct the original sequence.

Choices:

A:

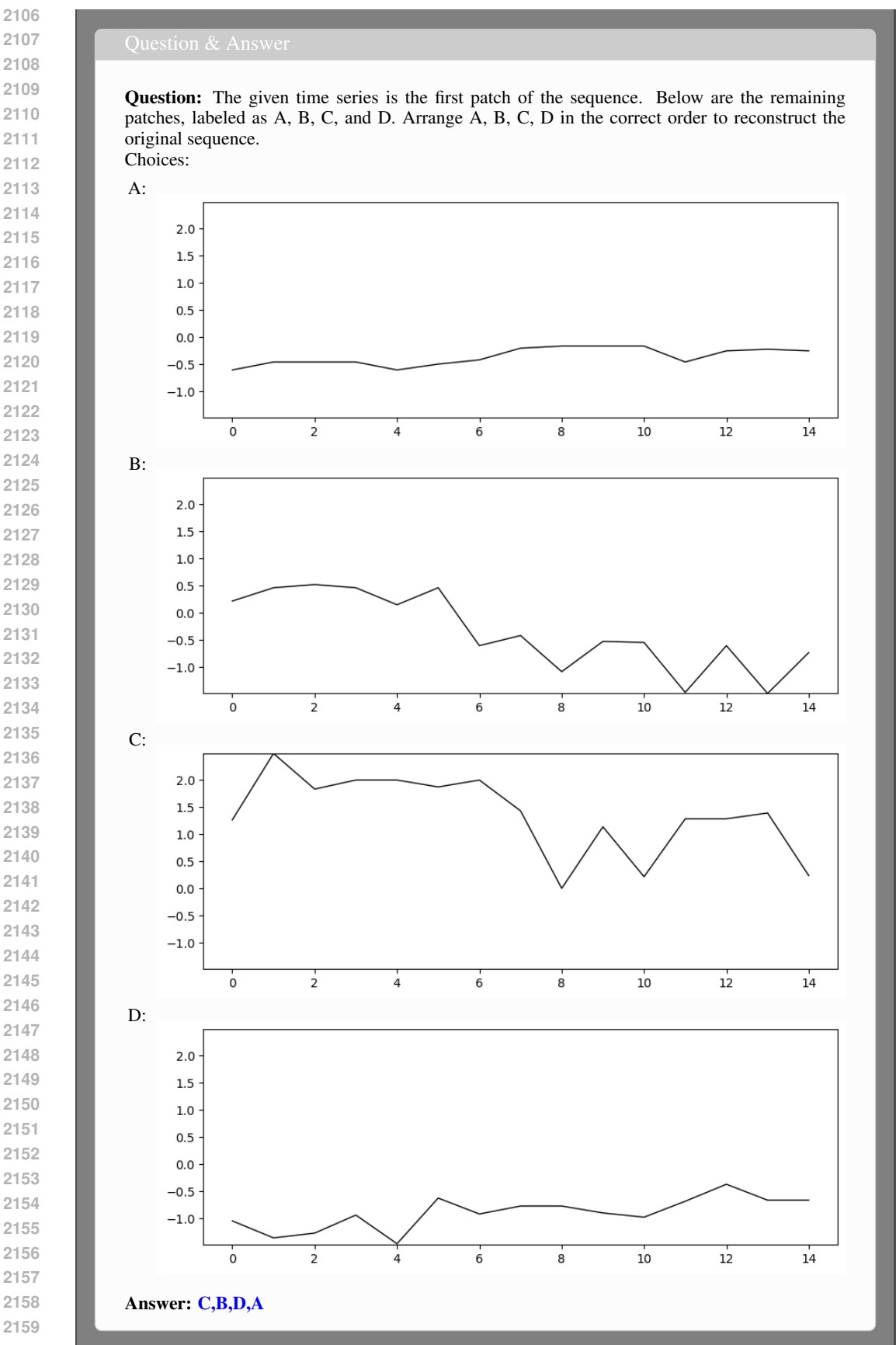

B:

C:

D:

**Answer: C,B,D,A**

**TRQA — Anomaly Detection Sample 1**

Time Series Info

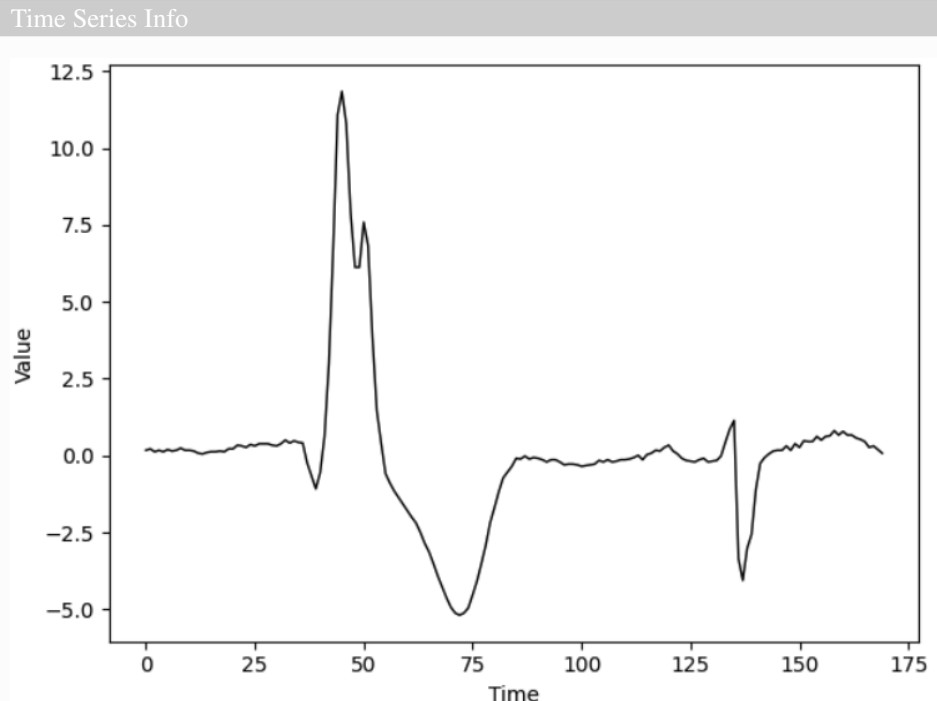

**Description:** This is an electrocardiogram (ECG) time series, and the anomalies represent ventricular premature contractions. The ECG recordings were made using Del Mar Avionics model 445 two-channel reel-to-reel Holter recorders, and the analog signals were recreated for digitization using a Del Mar Avionics model 660 playback unit. The digitization rate (360 samples per second per channel) was chosen to accommodate the use of simple digital notch filters to remove 60 Hz (mains frequency) interference.
**Question Type:** TF **Domain:** Healthcare **Dataset:** ECG

Question & Answer

**Question:** Determine whether the given time series contains anomalies. Respond ONLY with the letter of the correct choice (T or F).
Choices:

T: True.

F: False.

**Answer:** T

**TRQA — Classification Sample 1**

Time Series Info

**meta_info:** This time series comes from a dataset capturing process control measurements recorded by individual sensors during the fabrication of silicon wafers in semiconductor manufacturing, providing data for monitoring and classifying normal and abnormal production processes.
**Question Type:** MC **Domain:** manufacturing **Dataset:** UCR_Classification_Wafer

Question & Answer

**Question:** Classify the given time series into one of the categories below. Respond ONLY with the letter of the correct choice (A, B).
Choices:

 A: normal process

 B: abnormal process

**Answer: B**

# E EXPERIMENT ANALYSIS

We conducted an in-depth analysis of results from the selected Large Language Models. Specifically, our analysis is divided into two major categories: **Accuracy Correlate Analysis** and **Task-Specific Analysis**. For each analysis, we selected models from both commercial and open-source families. In particular, we chose the two best-performing models from Table 3 evaluated on our TRQA Benchmark—namely, GPT-4.1, Gemini 2.5 Flash, LLaMA3-8B, and Qwen3-8B. For LLaMA3.1-8B and Qwen3-8B, we analyzed both the zero-shot and instruction tuned models, resulting in a total of six models considered in our analysis.

## E.1 ACCURACY CORRELATE ANALYSIS

In this category, we examined how model accuracy or overall score correlates with input length.

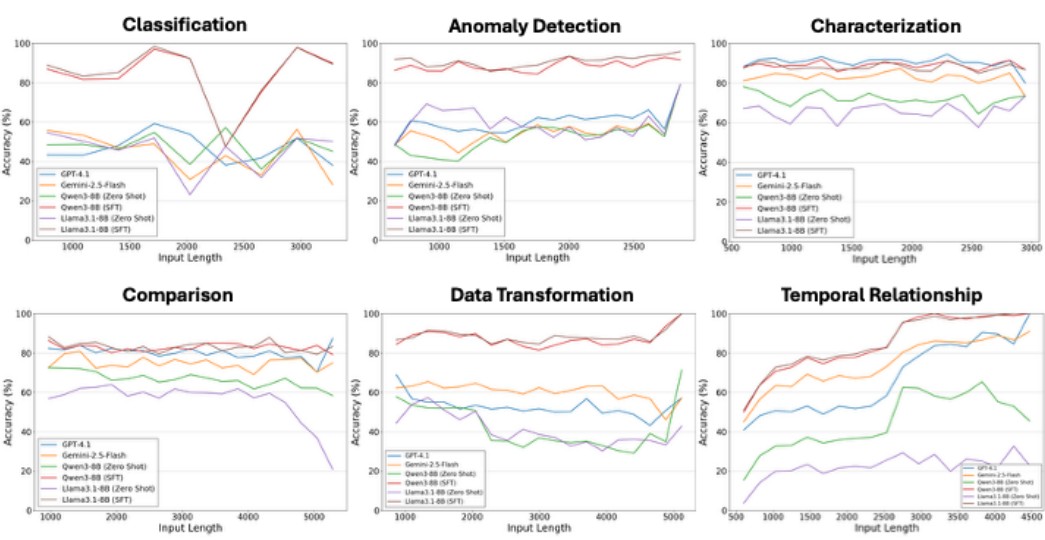

Figure 3: Input lengths vs. Accuracy by Tasks among six models.

**Input Length v.s. Accuracy**. To understand how input length impacts model accuracy, we conducted a detailed analysis comparing the length of each input with its corresponding accuracy. Specifically, the input length is calculated as *len(ts + description + domain + dataset + task + question_type + question)* with *String* type. The results are visualized in Figure 3. Each plot may contain input length starting and ending at different length as each task contains questions with different lengths. Across all six models and five tasks, excluding the Temporal Relation task, we observe a consistent trend that longer questions with greater input length generally result in lower accuracy and weaker overall model performance. However, the Temporal Relation task exhibits the opposite behavior, where accuracy improves with increasing input length. To understand this discrepancy, we conducted a detailed analysis of the four reasoning tasks *(Characterization, Comparison, Data Transformation, Temporal Relation)* in our proposed TRQA Benchmark, focusing on how different question types *(MC, TF, PZ)* and their corresponding input lengths correlate with model accuracy. The results are visualized in Figure 4. The results indicate that for all four reasoning tasks, MC and TF question types show a decline in accuracy with increasing input length, whereas the newly proposed PZ type exhibits the opposite trend. This implies that the model is actively using global contexts, all time series segments, to deduce the correct chronological order for answering PZ type question, which confirms that the model is engaging in deductive reasoning rather than local pattern matching. This proves that PZ type question is a rigorous probe for *Global Causal Reasoning*. Consequently, models whose accuracy improves with input length likely demonstrate a stronger ability to reason directly over time-series patterns.

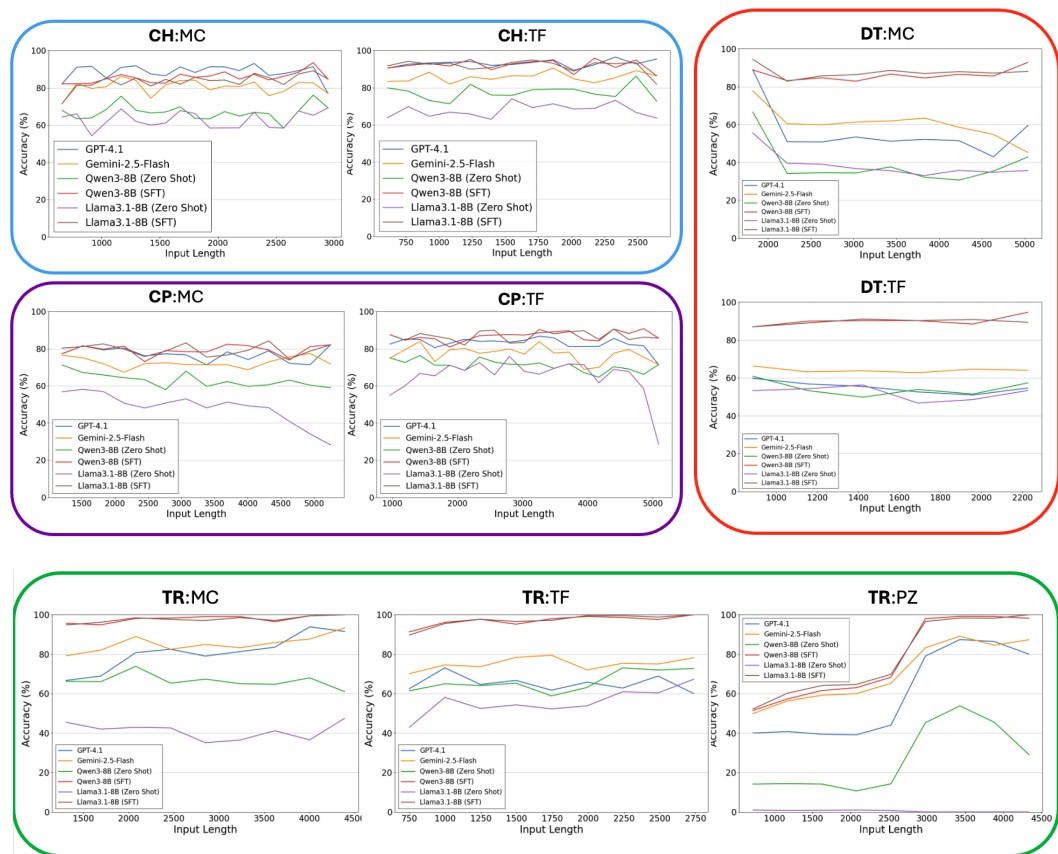

Figure 4: Input length vs. Accuracy by Question Types. CH, CP, DT, and TR denote Character-ization, Comparison, Data Transformation, and Temporal Relationship. MC, TF, and PZ denote true-or-false, multiple-choice, and puzzling.

## E.2 TASK SPECIFIC ANALYSIS

In this category, we examined how each model performed across the tasks proposed in our TRQA Benchmark. Specifically, we focused on the 3 reasoning tasks: Comparison, Data Transformation, and Temporal Relationship.

**Comparison**. We analyze model performance on the Comparison task, specifically investigating whether providing explicit domain-level context affects model accuracy. The task requires compar-ing two input time series, which we test under two conditions: (1) when both series originate from the same domain and (2) when they are from different domains. In both scenarios, the corresponding domain names are provided to the model as textual description. As shown in Table 8, we observe no significant performance difference between the same-domain and different-domain settings across either MC or TF questions. This finding suggests that our Comparison task is domain-invariant. Additionally, we also analyzed how question complexity affects GPT-4o's performance by varying the number of topics and subtopics used for generation. The model achieved an accuracy of 74.07% on questions derived from a single topic and subtopic, 72.17% for two, and 75.08% for three. These results indicate that the model's performance is notably stable, which again proves the quality of the proposed benchmark. Consequently, to answer correctly, models must reason based on the intrinsic patterns of the time series data itself, rather than relying on the textual context as a simple heuristic.

**Data Transformation**. We analyze model performance on the Data Transformation task, which is designed to evaluate a model's understanding of three transformation operators: Fourier Trans-

Table 8: Analysis of Comparison tasks.

| Group | Model | Same Domain | | Different Domain | |
|---|---|---|---|---|---|
| | | MC | TF | MC | TF |
| Zero Shot | GPT-4.1 | 76.27 | 83.62 | 78.06 | 83.48 |
| | Gemini-2.5-Flash | 70.97 | 77.90 | 74.06 | 77.63 |
| | Qwen3-8B | 62.99 | 70.67 | 63.54 | 71.60 |
| | LLaMA3.1-8B | 49.43 | 67.13 | 50.82 | 68.93 |
| Instruction Tuning | Qwen3-8B | 77.04 | 85.64 | 82.14 | 87.95 |
| | LLaMA3.1-8B | 78.02 | 86.32 | 81.24 | 87.35 |

form (FT), Wavelet Transform (WT), and First-Order Differencing (FOD). For each operator, we assess performance by measuring the accuracy on both MC and TF question formats. As shown in Table 9, for zero-shot evaluation, our key finding highlights a limitation in which both commercial and open-source models fail to provide accurate answers, except of FOD. In contrast, our instruction-tuned models show a better performance, achieving high accuracy across all tasks. However, FT is still very challenging even after instruction tuning. To explain our findings, we attribute this systematic performance disparity to two primary factors: the scope of temporal dependency and arithmetic complexity. As shown in Table 9, there is a clear performance degradation trend ($FOD > WT > FT$). This performance degradation is likely due to 3 reasons. (1) FOD relies solely on adjacent time steps ($x_t - x_{t-1}$), aligning well with the local attention capabilities of Transformers. (2) WT requires reasoning over localized windows in both time and frequency. As the dependency scope widens beyond immediate neighbors, model performance drops. (3) FT necessitates aggregating information from the entire sequence to determine frequency components. This global arithmetic reasoning is inherently challenging for LLMs' next-token prediction paradigm, resulting in the lowest performance. The results systematically validate that current LLMs struggle with tasks requiring global aggregation and complex arithmetic compared to robust local pattern matching, which also explains the results shown in Table 9.

Table 9: Analysis of Data Transformation Task. MC and TF denote multiple-choice and true-or-false, respectively. FT, WT, and FOD denote Fourier Transform, Wavelet Transform, and First-Order Differencing. We evaluate the accuracy on MC and TF questions from Data Transformation Task for each of the three transform operators.

| Group | Model | MC | | | TF | | |
|---|---|---|---|---|---|---|---|
| | | FT | WT | FOD | FT | WT | FOD |
| Zero Shot | GPT-4.1 | 26.32 | 35.39 | 91.90 | 51.36 | 51.64 | 59.81 |
| | Gemini-2.5-Flash | 27.97 | 53.19 | 100.00 | 50.25 | 53.59 | 85.90 |
| | Qwen3-8B | 9.06 | 28.40 | 66.4 | 52.57 | 52.05 | 52.66 |
| | LLaMA3.1-8B | 24.07 | 23.87 | 61.70 | 52.17 | 48.87 | 54.50 |
| Instruction Tuning | Qwen3-8B | 67.93 | 87.55 | 100.00 | 80.02 | 99.90 | 89.14 |
| | LLaMA3.1-8B | 71.83 | 88.79 | 99.70 | 82.54 | 89.24 | 98.36 |

**Temporal Relationship**. We analyzed model performance on the Temporal Relationship task, focusing specifically on our newly proposed Puzzling (PZ) question type. Beyond the input length versus accuracy analysis previously presented in Figure 4, we further examined how domain-level information influences model performance on Puzzling questions.

*Domain-Level Analysis.* The results are summarized in Table 10. The results show that the Web domain remains the most challenging for Puzzling questions across both zero-shot and instruction-tuning settings. Sales and Nature also exhibit lower accuracies, with Sales remaining difficult even after instruction-tuning. This indicates that domains such as Web and Sales impose greater temporal reasoning difficulty on models.

| Group | Model | Finance | Healthcare | Transport | Sales | Energy | Nature | Web |
|---|---|---|---|---|---|---|---|---|
| Zero Shot | GPT-4.1 | 62.22 | 57.75 | 55.86 | 52.62 | 52.53 | 48.87 | **46.53** |
| | Gemini-2.5-Flash | 76.59 | 80.12 | 76.65 | 66.54 | 76.95 | 72.46 | **63.51** |
| | Qwen3-8B | 27.81 | 27.76 | 24.36 | 22.87 | 24.32 | 21.43 | **17.90** |
| | LLaMA3.1-8B | **0.77** | 0.78 | 0.98 | 0.94 | 0.88 | 1.25 | 0.92 |
| Instruction Tuning | Qwen3-8B | 73.31 | 77.11 | 72.54 | 61.03 | 74.05 | 68.92 | **58.61** |
| | LLaMA3.1-8B | 75.25 | 77.50 | 72.22 | 61.80 | 75.80 | 71.16 | **60.86** |

Table 10: Domain v.s. Accuracy of the PZ question type in the Temporal Relationship task. The lowest and second-lowest results for each model are highlighted in **bold** and underlined, respectively.

## F  HUMAN EVALUATION

We further examine annotators' explanations in cases of disagreement. In the single-series benchmark, the largest source of mismatches is ambiguous questions (43%). Among well-formed cases, 24% involve trends, while volatility-, stability-, and periodicity-related issues each account for 10%. A small fraction (5%) reflects residual annotator uncertainty. In the multi-series benchmark, mismatches are more strongly tied to stochastic properties: volatility-related issues dominate (23%), followed by stability (13%). Periodicity- and lag-related issues each contribute 7%, while trend-related mismatches are rare (3%). Nearly half of the disagreements (47%) again arise from ambiguous questions, underscoring the greater interpretive difficulty of the multi-series setting. (See Figure 5)

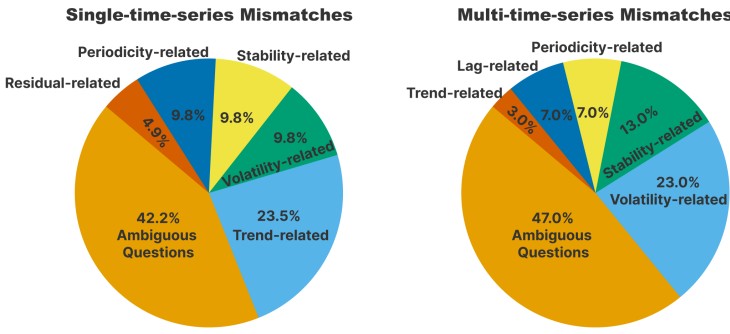

Figure 5: Human explanations for answer mismatches in TRQA

