# OpenReview forum: "TRQA: Time Series Reasoning Question And Answering Benchmark"
_ICLR.cc/2026/Conference — ICLR 2026 Conference Withdrawn Submission_

### Official Review · Reviewer_EP1r · 2025-10-29

**Soundness:** 2
**Presentation:** 2
**Contribution:** 2
**Rating:** 4
**Confidence:** 4

**Summary:**

A Q&A benchmark curated from existing domain specific time series.

**Strengths:**

1. The tasks are well defined.
2. Time series Q&A under domain specific context, i.e. combining time series understanding & domain specific knowledge is an interesting topic for the community.

**Weaknesses:**

1. The authors manually inspect a subset of questions and explain incorrect reasoning. The manuscript would further benefit from an empirical comparison. Specifically, reporting performance discrepancies between the filtered and retained Q&A pairs would more directly evidence how errors arise.

2. Evaluation would be stronger if it included program-aided or agentic LLM frameworks (e.g. [2]). Tasks such as comparison and data transformation are naturally suited to models with access to a code console, which could help isolate confounds (e.g. distinguishing misreading long numeric sequences from gaps in time-series reasoning).

3. The evaluation protocol should explicitly specify the input modality for time series (image vs. raw values/text). If charts/images are used, multivariate plots (e.g., 12-lead ECGs) can materially affect comprehension, so it is recommended to evaluate both modalities and analyze modality-specific performance differences.

3. From the appendix, some temporal-relation items appear under-specified without sufficient context. For example, in page 34 (“TRQA — Temporal Relationship Sample 1”), the rationale for preferring one continuation over another is unclear. I personally think that without explicit contextual priors, distinguishing continuations of a realized time series from a stochastic process may be ill-posed unless there are salient numerical or structural cues. In this case, what additional context or constraints (such as stationarity assumptions, domain priors) justify a unique “more likely” continuation in such items?

Relevant works:

1. https://arxiv.org/pdf/2410.14752
2. https://arxiv.org/abs/2410.04047

**Questions:**

Please see weaknesses.

---

### Official Review · Reviewer_zajU · 2025-10-30

**Soundness:** 3
**Presentation:** 3
**Contribution:** 2
**Rating:** 4
**Confidence:** 4

**Summary:**

The paper introduces TRQA, a large-scale benchmark for time-series question answering that spans many domains and task settings. It organizes six task families into two groups, namely conventional tasks such as anomaly detection and classification, and advanced tasks that probe representation characterization, cross-series comparison, transformation understanding, and temporal ordering. All items follow a unified QA interface with three formats, which are true or false, multiple choice, and a puzzling type that tests ordering and relational structure with position-based scoring. The benchmark defines clear construction rules for inputs and answers and provides standardized evaluation that supports repeatable comparisons. The study evaluates both commercial and open-source LLMs in zero-shot and instruction-tuned regimes and reports consistent gaps on structural reasoning and signal transformations. The stated goal is to move beyond pure forecasting or anomaly detection and to measure understanding and reasoning over time series in a way that is objective and reproducible.

**Strengths:**

- The benchmark spans thirteen domains and six complementary task types. It unifies inputs and outputs into a single QA protocol and supports three question formats that map to objective metrics. This enables fair and repeatable comparisons.

- TF and MC are scored by accuracy. PZ is scored by position-wise matches against ground truth, which is easy to reproduce and verify.

- The paper evaluates leading commercial LLMs and multiple open-source models with instruction tuning. The results highlight consistent gaps on structural and relational reasoning. PZ and transformation tasks remain challenging. The analysis of input length effects is informative and shows predictable degradation for longer inputs.

- The dataset construction process is clearly specified. It uses stratified sampling across domains and datasets, standardized preprocessing, and length control. This improves coverage and reduces hidden biases.

- Evaluation rules are simple and transparent. The absence of complex free-form grading reduces variance across replications and makes leaderboards easier to maintain.

**Weaknesses:**

1. Although the temporal relationship and transformation tasks are valuable, many items appear to be local or template-driven. The benchmark does not yet isolate multi-hop chains that require intermediate sub-conclusions within a single question. As a result, it is hard to quantify reasoning depth or track how errors accumulate across steps.

2. The paper does not label or control the number of operations required to answer a question. It also does not report performance as a function of step count or controlled statistical difficulty, beyond the broad input-length analysis. This limits interpretability of gains for different cognitive skills.

3. The experiments focus on LLMs, both commercial and open-source, together with instruction tuning on TRQA. The community has seen rapid progress in time-series specific foundation models and agent architectures. Without these baselines it is hard to position LLM-only performance against methods that are designed for numerical structure or external tool use.、

4. The main tables and summaries show overall and per-task numbers. The paper would benefit from qualitative cases and failure taxonomies that connect specific distractors, statistical conditions, or confusions to observed errors. This would help explain the differences among models and guide method design.

5. The motivation highlights the challenge of standardizing open-ended evaluation. The current design prioritizes objective scoring, which is reasonable, but it leaves a gap for controlled open-ended tasks that reflect real analysis workflows. A small sub-benchmark with automatic answer checking would improve coverage.

6.

**Questions:**

1. How does TRQA provide a distinct advantage over FutureX [1] and TSAIA [2] in scope, task design, and evaluation?

2. Can you annotate or synthesize questions with explicit multi-step chains, and then report accuracy as a function of the number of steps or operations per item?

3. Is it possible for you to include tasks that require integration across multiple long segments within the same question, and to provide curves that link context size and performance under controlled settings?

4. Can TRQA be adapted to time-series foundation models or agent frameworks with tool use, and are there any known obstacles to a direct comparison?

5. Would you consider publishing a small open-ended subset with automatic verification rules to complement TF, MC, and PZ?

[1] Zeng, Zhiyuan, et al. "Futurex: An advanced live benchmark for llm agents in future prediction." arXiv preprint arXiv:2508.11987 (2025).

[2] Ye, Wen, et al. "When LLM Meets Time Series: Can LLMs Perform Multi-Step Time Series Reasoning and Inference." arXiv preprint arXiv:2509.01822 (2025).

---

### Official Review · Reviewer_Tg1y · 2025-10-30

**Soundness:** 2
**Presentation:** 3
**Contribution:** 2
**Rating:** 2
**Confidence:** 3

**Summary:**

This paper introduces TRQA, a benchmark for time-series question answering that unifies six tasks—anomaly detection, classification, characterization, comparison, data transformation, and temporal relationship reasoning—into a QA format across 13 domains. While the dataset is large and carefully documented, the conceptual novelty is limited. Similar formulations already appear in Time-MQA and ChatTS, and TRQA primarily aggregates existing ideas rather than providing new methodological insights.

A major concern is the synthetic generation pipeline: most questions and answers are produced and verified by large language models, creating potential circularity between benchmark construction and evaluation. Human validation covers only about 0.3 % of the data, leaving quality assurance weak. The evaluation focuses solely on LLMs, without comparing to non-LLM or numerical baselines, so it is unclear whether the benchmark tests temporal reasoning or text comprehension.

Despite clear writing and strong reproducibility, TRQA offers little analysis of model failure types or reasoning errors. Overall, this work may be valuable as a dataset release but lacks sufficient conceptual and empirical depth for ICLR acceptance.
Overall, TRQA is an interesting benchmark that organizes diverse time-series reasoning tasks into a unified QA framework. However, its methodological novelty is limited, the dataset quality depends heavily on self-referential LLM generation, and the evaluation lacks critical analysis.

**Strengths:**

Pros:
1. comprehensive coverage of tasks.
The authors attempt to move beyond conventional forecasting and anomaly detection benchmarks, covering a wider range of reasoning abilities (comparison, transformation, temporal ordering).
Detailed documentation and reproducibility.
2. the paper provides extensive appendices (B–E) with dataset descriptions, templates, and human validation details. This level of transparency is commendable.
3. Inclusion of “puzzling” (PZ) questions.
While not novel per se, the PZ format introduces a valuable diagnostic dimension related to temporal relational reasoning.

**Weaknesses:**

Cons:
1. Limited novelty and conceptual contribution.
The core idea—recasting standard time-series tasks into QA form—is not new.
Prior works such as Time-MQA (Kong et al., 2025) and ChatTS (Xie et al., 2025) already explored similar formulations.
TRQA’s main addition is the aggregation of multiple task types, but it does not present new methodological insights, metrics, or reasoning frameworks.
As such, the work feels more like an incremental benchmark collection than a conceptual advancement.

2. Synthetic generation pipeline raises validity concerns.
The benchmark heavily relies on LLM-generated questions and answers, then uses other LLMs to verify them.
Despite the multi-model “consensus,” this design introduces circularity — the same class of models used to produce the benchmark are later evaluated on it.
This weakens the claim that TRQA objectively measures reasoning rather than memorized or stylistic biases of GPT-family models.

3. Human validation is too shallow.
Only 600 samples (out of 210k) are manually checked by six annotators.
This 0.3% coverage is insufficient to ensure quality and diversity of reasoning types, particularly when most QA pairs are synthetic.
The reported 91% agreement rate does not adequately capture inter-annotator variance or error typology.

4. Evaluation design lacks meaningful baselines.
The experiments focus exclusively on LLMs (GPT-4, Gemini, LLaMA3, Qwen, etc.), ignoring non-LLM time-series models (e.g., Chronos, TimeMoE, TSMixer).
Without these comparisons, it’s unclear if TRQA tests time-series reasoning or simply language understanding of templated text.
5. Puzzling (PZ) task unclear in utility. The paper claims PZ reflects human-like temporal cognition, but provides no analysis of what model errors in this setting actually mean. Accuracy on “ordering four shuffled patches” seems arbitrary and may not reflect reasoning ability, especially since the questions are generated synthetically.
6. Limited insight beyond performance tables.
The results largely restate numeric scores (e.g., Table 3) without deeper analysis.

**Questions:**

1. Could you elaborate on how you prevent data leakage or circularity in the LLM-generated QA pairs? For instance, are any of the evaluated models (e.g., GPT-4, Gemini, Claude) used directly in question generation or consensus labeling? If so, how is independence ensured during evaluation?

2. Only 600 samples (≈0.3%) are manually validated. Do you have plans to extend human auditing or provide inter-annotator agreement statistics per task (e.g., for characterization vs. comparison)? Could stronger human supervision change benchmark reliability?

3.Why are no non-LLM baselines (e.g., Chronos, TimeMoE, or TSMixer) included? Would the authors consider adding classical or foundation-model baselines to demonstrate that TRQA truly measures temporal reasoning rather than language pattern recognition?

4. Beyond accuracy, have you analyzed qualitative failure cases? For example, what specific reasoning patterns or temporal dependencies do models consistently miss (e.g., trend direction, seasonality, ordering)? This could greatly strengthen the claim that TRQA probes “reasoning” rather than surface similarity.

---

### Official Review · Reviewer_5K4J · 2025-11-02

**Soundness:** 2
**Presentation:** 2
**Contribution:** 2
**Rating:** 2
**Confidence:** 5

**Summary:**

The paper proposes a large time series reasoning and questioning benchmark covering 13 different domains. It comprises of two kinds of tasks including conventional problem such as classification and anomaly detection, and reasoning tasks such as comparison and characterization. The questions are expressed in the form of multiple choice, true / false, and "puzzling"-type questions. Each question in the benchmark comprises of one of more time series, along with a question, answer, and contextual information about the time series.

**Strengths:**

The paper addresses the important and pressing need for real-world datasets to evaluate large language models, and foundation models' reasoning capabilities.

**Weaknesses:**

- **Writing:** I would recommend that the authors add descriptive captions to their figures and tables, that tell a story. For example, the caption in Table 1 does not example what # reason task corresponds to. It would also help the reader, if the authors wrote down something like "We introduce TRQA, a novel large-scale benchmark comprising 210k samples across 13 domains, covering 6 tasks and 3 types of questions." in the caption to highlight their contribution.

> Mtbench (Williams et al., 2025) proposes a QA benchmark mainly for forecasting tasks.
- The cited benchmark is called Context is Key, and I would not count it as a QA benchmark.

- **Lack of discussion and comparison with prior work:** The paper lacks discussion of multiple studies in the intersection of time series, language and question-answering, for example JoLT [1] which solves the problem of time series captioning & question answering in the clinical context, and TimeSeriesExam [2], which proposed a scalable benchmark with synthetic data to evaluate LLM's understanding of time series data, and also comprises of many categories of questions proposed in the study, e.g., comparison-type, anomaly detection, etc. Prior work by Fons et al. [3] and Zhang et al. [4] are also potentially relevant. On the other hand, **some cited work is not relevant**, for example "Unseentimeqa: Time-sensitive question-answering beyond llms’ memorization" has nothing to do with time series data (based on a cursory read).

- **Missing details:** The paper would benefit from adding some missing details. See the section on Questions for more details.

- **Some settings are not reflective of the real-world*: For example, for anomaly detection questions the authors "downsample the normal samples to balance the classes at a 1:1 ratio". This defeats the purpose of anomaly detection where anomalies are far and few. The time series are also z-score normalized, whereas real-world time series would seldom be standardized.

- **Novelty of anomaly detection and classification questions:** I would like to better understand how these questions are normal. There are many existing datasets such TSB-UAD, UCR, and UCR/UEA benchmarks for anomaly detection and classification respectively. Is the contribution converting anomaly detection datasets to the Q&A format?

- **Issues with evaluation:** LLMs are known to prefer text generated by themselves, over text generated by other models. How do we know that is not the case in this benchmark? The questions were generated using GPT-4, and it is also the best performing model.

- **Puzzle-type questions:** These questions are a contribution of this study. I would like to better understand what these questions are evaluating, how are they novel and valuable?


### References
1. Cai, Yifu, et al. "Jolt: Jointly learned representations of language and time-series." Deep Generative Models for Health Workshop NeurIPS 2023. 2023.
2. Cai, Yifu, et al. "Timeseriesexam: A time series understanding exam." arXiv preprint arXiv:2410.14752 (2024).
3. Fons, Elizabeth, et al. "TADACap: Time-series Adaptive Domain-Aware Captioning." Proceedings of the 5th ACM International Conference on AI in Finance. 2024.
4. Zhang, Xiyuan, et al. "Does Multimodality Lead to Better Time Series Forecasting?." arXiv preprint arXiv:2506.21611 (2025).

**Questions:**

> Unless otherwise specified, all samples have a random length in [32, 256], and are z-scored to reduce data bias.
-What kind of bias are you referring to? Does normalizing time series not make the benchmark unrealistic? Real-world time series do not come standardized.

- **Temporal relationship questions:** I would like to better understand what these types of questions are. Should I think about them as a forecasting problem in the form of a MCQ / True and False question? What are these questions testing?

- Can you provide more details on how you instruction tuned the models?

> Given the anchor x, we construct a set of 10 comparison samples with the same length as x
What are the characteristics of time series that the model comparison?

- Lotsa (Woo et al., 2024), Time-300B (Shi et al., 2024), and UTSD (Ma et al., 2024)-- Why were these datasets used? LOTSA and Time-300B, to the best of my knowledge, only comprise of datasets which can be used for forecasting as they were used to pre-train forecasting foundation models. On the other hand, I did not find that UTSD released any dataset (based on my cursory reading), and the datasets used in the study seem very similar to Time Series Pile proposed by [1].

> Six Ph.D.-level experts manually annotate 600 questions (300 each),
- What is the inter-rater agreement? What instructions were they provided?

- How was time series tokenized and passed to the models?

> our key finding highlights a limitation in which both commercial and open-source models fail to provide accurate answers, except of FOD.
- Is there a systematic reason why this is the case?


###
1. Goswami, Mononito, et al. "Moment: A family of open time-series foundation models." arXiv preprint arXiv:2402.03885 (2024).

---

### Note · Authors · 2025-12-02

I have read and agree with the venue's withdrawal policy on behalf of myself and my co-authors.